# FairTune: Optimizing Parameter Efficient Fine Tuning for Fairness in Medical Image Analysis

**Raman Dutt**[1], **Ondrej Bohdal**[1], **Sotirios A. Tsaftaris**[1], **Timothy Hospedales**[1,2]
[1]The University of Edinburgh  [2]Samsung AI Center, Cambridge
{raman.dutt,ondrej.bohdal,s.tsaftaris,t.hospedales}@ed.ac.uk

## Abstract

Training models with robust group fairness properties is crucial in ethically sensitive application areas such as medical diagnosis. Despite the growing body of work aiming to minimise demographic bias in AI, this problem remains challenging. A key reason for this challenge is the fairness generalisation gap: High-capacity deep learning models can fit all training data nearly perfectly, and thus also exhibit perfect fairness during training. In this case, bias emerges only during testing when generalisation performance differs across subgroups. This motivates us to take a bi-level optimisation perspective on fair learning: Optimising the learning strategy based on validation fairness. Specifically, we consider the highly effective workflow of adapting pre-trained models to downstream medical imaging tasks using parameter-efficient fine-tuning (PEFT) techniques. There is a trade-off between updating more parameters, enabling a better fit to the task of interest vs. fewer parameters, potentially reducing the generalisation gap. To manage this tradeoff, we propose *FairTune*, a framework to optimise the choice of PEFT parameters with respect to fairness. We demonstrate empirically that *FairTune* leads to improved fairness on a range of medical imaging datasets. The code is available at https://github.com/Raman1121/FairTune.

## 1 Introduction

The use of AI in healthcare applications is growing rapidly. Powerful new models enabled by large datasets (Mei et al., 2022; Ghesu et al., 2022; Irvin et al., 2019) are rapidly being developed, leading to highly performant automated diagnosis systems (Tiu et al., 2022) that are increasingly being deployed clinically in clinical practice (Esteva et al., 2021; Dutt et al., 2022; Vats et al., 2022). However, AI models have repeatedly been shown to exhibit unwanted biases towards various demographic subgroups (Seyyed-Kalantari et al., 2021; Obermeyer et al., 2019; Larrazabal et al., 2020; Ricci Lara et al., 2022) – for example by providing substantially worse performance on disadvantaged subgroups defined by protected attributes such as gender, race, age, and socioeconomic status. This is obviously socially, ethically, and clinically problematic, especially in potentially life-and-death situations that arise in healthcare.

The issue of biased and inequitable AI systems has prompted a growing body of research striving to analyze the origins of bias and develop interventions to mitigate model bias (Xu et al., 2023). Nevertheless, recent investigations cast doubt on the extent of progress achieved thus far. Notably, Zietlow et al. (2022) postulate that the majority of existing interventions aimed at promoting fairness prove ineffective when applied to deep models, which are commonly utilized for tasks involving images and text data. The reason behind this ineffectiveness lies in the nature of these interventions, such as those proposed by Sagawa et al. (2020) and Zhao et al. (2019), which impose constraints on the *training data*. For instance, they enforce equal performance across subgroups (Zhao et al., 2019). However, while such constraints can impact the training of shallow models typically employed for tabular data, deep models possess the capability to perfectly fit all training data, rendering these fairness constraints automatically satisfied and devoid of any influence on the model's learning process. We substantiate this well-documented challenge empirically in Figure 1, which illustrates that, in a typical medical image analysis scenario, the training data can be fitted flawlessly. Consequently, the model is already *intrinsically equitable within the training set*. The observed bias in real-world applications emerges during testing, primarily due to differential generalization across subgroups.

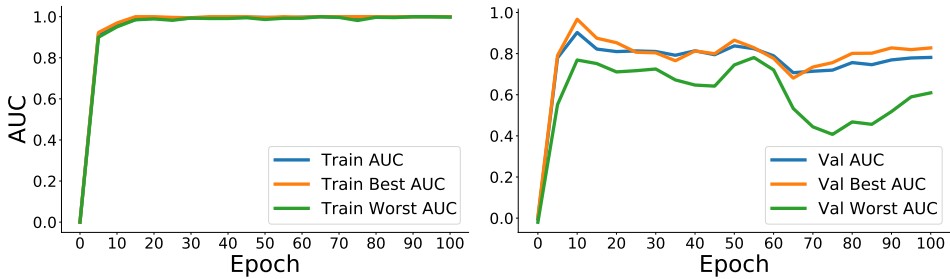

Figure 1: Bias arises during train-test generalisation. Left (Training AUROC): High-capacity deep models can exhibit perfect group fairness during training because they can classify all the training data perfectly. Right (Validation AUROC): Bias arises because the disadvantaged subgroup has worse generalisation error than the privileged subgroup. Fine-tuning ViT-Base on the Papila dataset.

Another recent study (Zong et al., 2023) empirically evaluated a wide range of fairness interventions designed to regularise deep model learning on a large suite of medical image analysis tasks. However, they found that prior progress was over-estimated. When subjected to a standardized hyperparameter tuning procedure for a fair evaluation, none of the existing fairness interventions exhibited a statistically significant enhancement in fair learning when compared to the conventional approach of supervised learning by empirical risk minimization (ERM).

In this research paper, we introduce a novel approach to fair learning that addresses the challenge highlighted by Zietlow et al. (2022) and depicted in Figure 1. Our method is rooted in the concept of capacity control, and involves introducing a form of regularization during the learning process specifically tailored to *minimize bias in unseen data*. To accomplish this, we operate within the pre-train/fine-tune framework (Mei et al., 2022; Yosinski et al., 2014; Tang et al., 2022; Zong et al., 2023). This framework entails initializing models through pre-training on extensive external datasets like ImageNet (Deng et al., 2009), followed by fine-tuning on comparatively smaller medical imaging datasets. In this context, as we progressively update the model from its initial pre-trained state, the risk of overfitting to the nuances of the training set increases, leading to the generalization gap illustrated in Figure 1. Hence, the primary challenge lies in restraining the extent of model updates. In this regard, we will illustrate that employing parameter-efficient fine-tuning techniques, which involve the selective updating of a subset of network parameters (Dutt et al., 2023), can result in more equitable generalization. However, this approach poses a critical question: *"Which parameters should be updated to maximize fairness?"* To tackle this question, we introduce our framework named **FairTune**, designed to search for the optimal parameter update mask. We seek the mask that, when applied to constrain the fine-tuning process, yields a high degree of fairness in the validation data. Our empirical findings consistently demonstrate that FairTune outperforms Empirical Risk Minimization (ERM) in terms of fairness across various medical imaging benchmarks.

To summarise our contributions: **(1)** We directly corroborate the conjecture of Zietlow et al. (2022) that bias arises during train-test generalisation (Figure 1). **(2)** In contrast to existing fairness interventions, we introduce a new fair learning approach that regularises learning so as to optimise validation fairness (cf: existing methods that ineffectively target training fairness). **(3)** Our empirical findings across a diverse set of benchmarks consistently demonstrate that **FairTune** reliably improves performance over ERM.

## 2 RELATED WORK

### 2.1 FAIRNESS IN MEDICINE

Bias and unfairness have been widely reported in biomedical AI (Seyyed-Kalantari et al., 2021; Ricci Lara et al., 2022; Obermeyer et al., 2019). Biases can arise from a complex array of different underlying causes including dataset imbalance, label noise, and reliance on underlying spurious correlations. A particularly problematic manifestation is that of bias amplification (Lloyd, 2018; Hall et al., 2022), where biases that exist in the training set are amplified by the model's predictions during deployment. Measuring fairness is itself a complex problem, as many different fairness met-

rics have been proposed, with no consensus on a single preferred metric. For example, optimising for equal performance among demographic subgroups (Dwork et al., 2012; Verma & Rubin, 2018) is intuitive. But this can lead to the *levelling down* phenomenon (Zietlow et al., 2022), where fairness is achieved by decreasing the performance of the advantaged group to match the disadvantaged group – potentially even including pathological solutions of reducing both groups' performance to zero. Achieving fairness by levelling down has been criticised as violating the ethical principles of beneficence and non-maleficence (Beauchamp, 2003; Chen et al., 2018; Ustun et al., 2019). We also remark that evaluating systems for fairness is itself complex (Zong et al., 2023; Verma & Rubin, 2018) as fair learning is inevitably a multi-objective problem that seeks to simultaneously achieve potentially conflicting goals of good overall performance and good fairness.

## 2.2 PREVIOUS ATTEMPTS TO SOLVE FAIRNESS

Fair machine learning has now been widely studied, with numerous methods being proposed that address bias reduction via both pre-processing (e.g., data re-balancing) and post-processing, as well as interventions aimed at guiding the learning algorithm to generate a fairer predictor. Due to the large volume of the proposed methods in the literature, we refer the readers to comprehensive surveys (Mehrabi et al., 2021; Caton & Haas, 2023; Zong et al., 2023) for a more in-depth exploration of the available techniques and their nuances. A crucial observation, however, is that a large family of methods (Sagawa et al., 2020; Zhao et al., 2019; Agarwal et al., 2018b; Beutel et al., 2017; Diana et al., 2021; Jeong et al., 2023; Donahue et al., 2016; Yii et al., 2022; Donini et al., 2018; Dumoulin et al., 2016; Kim et al., 2019; Kleindessner et al., 2022; Lohaus et al., 2020; Martinez et al., 2020; Padala & Gujar, 2020; Wang et al., 2020; Zafar et al., 2017; Wu et al., 2022; Park et al., 2022) rely on imposing fairness constraints on the *training* set. As suggested by Zietlow et al. (2022), these are ineffective in the deep learning regime where constraints are trivially satisfied by a classifier that achieves 100% training accuracy (Figure 1). Another family of methods endeavours to introduce various forms of regularization during model training, aiming to enhance generalization, such as achieving domain independence. While some of these studies initially reported promising outcomes, a recent exhaustive benchmarking study (Zong et al., 2023) has indicated that these assertions were premature. When evaluated across multiple benchmarks, existing methods consistently fall short of systematically outperforming a well-tuned supervised learning baseline for fairness (ERM).

We are inspired by studies such as Zietlow et al. (2022); Zong et al. (2023) to design an algorithm that tunes how to regularise learning with the explicit objective of optimising for *validation fairness*.

## 2.3 PARAMETER-EFFICIENT FINE-TUNING

Fine-tuning models that have been pre-trained on large datasets is common practice in deep learning (Yosinski et al., 2014; Kornblith et al., 2019). Leveraging a pre-trained initialization enables downstream tasks to be learned with significantly less data compared to training from scratch. Parameter-Efficient Fine-Tuning (PEFT) methods are a family of techniques geared towards improving the fine-tuning process. They achieve this by carefully selecting a small subset of parameters for updating during fine-tuning while keeping the majority frozen. The underlying concept is that this judicious choice of selective updates should facilitate effective adaptation to the target task (via the minority of updatable parameters) while guarding against overfitting (courtesy of the majority of frozen parameters). A growing number of PEFT methodologies have emerged, each distinguishing itself by its specific selection of parameters for updating. These selections may include biases (Ben Zaken et al., 2022), attention matrices (Touvron et al., 2022), or normalization layers (Basu et al., 2023). Alternatively, some methods introduce and learn specific sets of new parameters, such as low-rank adapters (Hu et al., 2022), all while maintaining the entire pre-trained backbone in a frozen state. PEFT techniques have gained wide popularity in mainstream NLP and computer vision applications, although their adoption in medical image analysis tasks remains nascent (Dutt et al., 2023; Ma & Wang, 2023; Wu et al., 2023; Zhang & Liu, 2023).

In this work, we aim to demonstrate that PEFT (Parameter-Efficient Fine-Tuning) offers benefits beyond enhancing traditional generalization capabilities. Specifically, our findings will illustrate that PEFT can enhance fairness by narrowing the generalization gap, especially for disadvantaged subgroups, as depicted in Figure 1. Nevertheless, a central challenge persists across all existing PEFT methods, namely, they rely on heuristic approaches for partitioning parameters into frozen

and updatable sets. Current methods do not offer a principled or learned method for establishing the optimal partition. This becomes particularly crucial, because the ideal PEFT assumption, i.e., the freeze/update partition, may be dataset dependent. For instance, larger datasets might accommodate a more extensive parameter update without suffering from overfitting compared to smaller datasets. The key novelty of this paper lies in our approach: instead of prescribing a specific PEFT update mask, we introduce a framework designed to autonomously determine the optimal PEFT mask that maximises validation fairness.

## 3 METHODOLOGY

### 3.1 FAIRNESS METRICS

We focus on evaluating the fairness of binary classification of medical images. Given an image $x$ we predict its diagnosis label $y$ in a way that aims to be independent of any sensitive attribute $s$ (age, sex, ethnicity, etc.) so that the trained model is fair and does not unduly disadvantage any particular demographic subgroup. There are a plethora of metrics to measure fairness such as equality of opportunity, equal odds, subgroup performance difference, and so on (Verma & Rubin, 2018). Each of these may be more appropriate for different social and economic situations. Our overall framework is agnostic to the choice of fairness metric used, as our contribution is an approach to optimise for any user-specified fairness metric. However, for most of our experiments, we will optimise the metric of most-disadvantaged group performance (Sagawa et al., 2020). In this setting we are given a loss function $\mathcal{L}(\mathcal{D}; \theta)$ (e.g., cross-entropy, or 1 - area under ROC curve) for model $\theta$ on dataset $\mathcal{D}$. We assume it can be evaluated for different subgroups $s$ of the dataset $\mathcal{D}$ as $\mathcal{L}(\mathcal{D}_s; \theta)$. Then the metric for fair learning is

$$\mathcal{L}^{fair} = \max_{s \in S} \mathcal{L}(\mathcal{D}_s; \theta). \tag{1}$$

We will also report other metrics such as the fairness *gap*, estimated as the performance difference between the disadvantaged and privileged subgroups, $(\max_s \mathcal{L}(\mathcal{D}_s; \theta) - \min_s \mathcal{L}(\mathcal{D}_s; \theta))$.

### 3.2 PARAMETER-EFFICIENT FINE-TUNING

In PEFT, we fine-tune only a subset of parameters $\phi \subset \theta$ such that $|\phi| \ll |\theta|$. PEFT strategies can be interpreted as specifying a sparse binary mask $\omega$ that determines what parts of $\theta$ should be updated. Given parameters $\theta_0$ of the pre-trained model and a change $\Delta\phi$ to be applied to their values, the fine-tuning process can be described as

$$\Delta\phi^* = \arg\min_{\Delta\phi} \mathcal{L}^{base}\left(\mathcal{D}^{train}; \theta_0 + \omega \odot \Delta\phi\right).$$

where $\mathcal{L}^{base}$ is a standard deep learning loss such as cross-entropy for classification.

Different PEFT methods essentially correspond to different structures on the sparsity structure of the binary mask $\omega$. For example, BitFit (Ben Zaken et al., 2022) solely updates bias parameters in a neural network. Attention Tuning (Touvron et al., 2022) enables updating all the attention matrices in a transformer, and so on. These methods are generally effective in reducing overfitting when learning large models on small datasets thanks to eliminating most parameter updates. We will show that they are also effective in improving generalisation fairness compared to conventional fine-tuning.

There are two key outstanding challenges, however: (1) The optimal PEFT strategy (binary mask $\omega$) is dataset-dependent. For example, a sparser mask $\omega$ may be preferred for a smaller target task with greater risk of overfitting, and a denser mask may be preferred for a task that is more different to the pre-training task and thus requires stronger adaptation. (2) The optimal PEFT strategy may depend on the ultimate generalisation objective. For example, a sparser mask $\omega$ might be preferred for fair generalisation compared to conventional overall generalisation. We present a solution to both of these issues by introducing an algorithm to optimise the mask $\omega$ with respect to a fair generalisation objective.

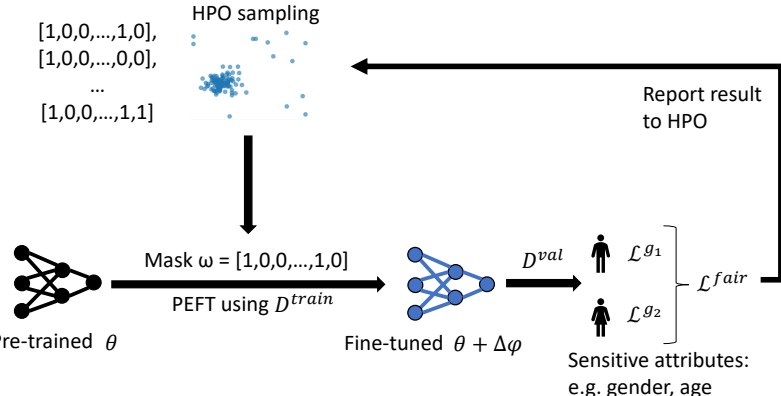

Figure 2: Illustration that shows how our approach optimises the structure of PEFT with respect to fairness. Hyperparameter optimisation (HPO) selects a mask that decides which components of a pre-trained model $\theta$ are fine-tuned using PEFT. For each sampled mask, the fine-tuned model is evaluated on the validation set to compute the fairness loss $\mathcal{L}^{fair}$, which is then reported to the HPO algorithm that decides what masks to sample and which is the final best option.

### 3.3 Optimising PEFT for Fairness

We begin with a pre-trained model $\theta_0$, a dataset ($\mathcal{D}$) split into training, validation and test sets ($\mathcal{D}^{train}, \mathcal{D}^{val}, \mathcal{D}^{test}$). Each dataset $\mathcal{D} = (\mathcal{X}, \mathcal{Y}, \mathcal{S})$ contains a set of images $\mathcal{X}$, labels $\mathcal{Y}$ and sensitive attribute metadata $\mathcal{S}$. We also define a search space for PEFT masks $\omega \in \Omega$. The goal is to find $\omega$ that leads to the best fair generalisation (Sec 3.1) when conducting PEFT learning (Sec 3.2).

**Bi-level Optimization (BLO):** We formalize our problem statement as a bi-level optimization problem consisting of an inner and an outer loop. In the inner loop, we fine-tune the pre-trained model on the medical dataset ($\mathcal{D}^{train}$) using a conventional loss $\mathcal{L}^{base}$ and PEFT mask $\omega$. In the outer loop, we search for the PEFT mask $\omega$ which leads the inner loop to produce the fairest outcome on the validation set ($\mathcal{D}^{val}$), as measured by $\mathcal{L}^{fair}$. More formally, we solve Equation 2.

$$\omega^* = \underset{\omega}{\arg\min} \mathcal{L}^{fair}\left(\mathcal{D}^{val}; \Delta\phi^*\right)$$
$$\text{such that} \quad \Delta\phi^* = \underset{\Delta\phi}{\arg\min} \mathcal{L}^{base}\left(\mathcal{D}^{train}; \theta_0 + \omega \odot \Delta\phi\right). \tag{2}$$

There are a number of possible strategies for solving BLO problems such as Equation 2 including meta-gradient, evolutionary search, Bayesian Optimisation and others (Hospedales et al., 2021; Sinha et al., 2018; Liu et al., 2021). In practice, we adopt a hybrid approach with a gradient-free Tree-structured Parzen Estimator (TPE) (Bergstra et al., 2011) with successive halving (SH) strategy (Jamieson & Talwalkar, 2016) for optimising $\omega^*$ in the outer loop (Akiba et al., 2019), and conventional long-horizon gradient-descent fine-tuning in the inner loop. We illustrate the process in Figure 2 and provide full details in Algorithm 1. Additional details on the HPO are given in Appendix A.5

Besides the selective-update mask $\omega$, the learning rate $\alpha$ also provides a coarse cue of how much to update. For example a suitably curtailed learning rate would prevent the most egregarious overfitting shown in Figure 1. We also optimise $\alpha$ along with $\omega$ within the same HPO process of Algorithm 1.

## 4 Experiments

### 4.1 Experimental Setup

**Architectures:** Our experiments adopted the Vision Transformer (ViT) implementation present in the *Pytorch Image Models* package (Wightman, 2019). A ViT consists of several blocks and each block contains two normalization layers (LN1 and LN2), Multi-Head Self-Attention (MHSA) sub-block and MLP (MLP) sub-components. The normalization layers (LayerNormalization Ba et al.

---

**Algorithm 1** Optimizing PEFT for fairness

---

1: **Input:** pre-trained model $\theta_0$, $\alpha$: fine-tuning learning rate, number of trials $T_N$
2: **Output:** fine-tuned model $\theta_0 + \omega \odot \Delta\phi$ and mask $\omega$
3: **while** number of completed trials $< T_N$ **do**
4:     Initialize $\Delta\phi = 0$ for $\phi \leftarrow \phi_0 \subset \theta_0$
5:     Propose mask $\omega \leftarrow HPO$
6:     **while** not converged **do**
7:         $\Delta\phi \leftarrow \Delta\phi - \alpha\nabla_\phi\mathcal{L}^{base}\left(\mathcal{D}^{train}; \theta_0 + \omega \odot \Delta\phi\right)$     // PEFT
8:     **end while**
9:     Evaluate $\mathcal{L}^{fair}(\mathcal{D}^{val}; \theta_0 + \omega \odot \Delta\phi)$ and report to the HPO algorithm
10: **end while**
11: Return the best (most fair) mask $\omega$ and fine-tuned model $\theta_0 + \omega \odot \Delta\phi$

---

(2016)) are present before and after MHSA. The MLP consists of two fully-connected layers with GELU non-linearity in between. The base variant of ViT consists of 12 such blocks.

**Baselines:** We compare our approach with: (1) full training from scratch. (2) conventional full fine-tuning, as conducted in (Zong et al., 2023) (where it is referred to as ERM) where every layer of an ImageNet pre-trained model is adapted on the medical image task, (3) linear readout, where the ImageNet pre-trained feature extractor is frozen and only the classification head is learned, as conducted in (Azizi et al., 2021; Chen et al., 2020), (4) PEFT method Attention Tuning (Touvron et al., 2022) where only attention matrices are fine-tuned, (5) PEFT method Layer Norm Tuning (Basu et al., 2023) where only layer-norm parameters are fine-tuned, (6) FairPrune (Wu et al., 2022), a method that achieves fairness by pruning the model parameters post-training, and (7) Fair Supervised Contrastive Loss (FSCL) (Park et al., 2022) that inherits the properties of supervised contrastive learning and penalizes the usage of sensitive attribute information in representation for improving fairness.

We remark that the thorough benchmark in (Zong et al., 2023) already dismissed a suite of algorithms designed for the purpose of fair learning as equal or worse than ERM/Fine-Tuning (Vapnik, 1999), including **DomainInd** Wang et al. (2020), **LAFTR** Madras et al. (2018), **CFair** Zhao et al. (2019), **LNL** Kim et al. (2019), **EnD** Tartaglione et al. (2021), **ODR** Sarhan et al. (2020), **Group-DRO** Sagawa et al. (2020), **SWaD** Cha et al. (2021), and **SAM** Foret et al. (2021).

**Search Space:** We define a PEFT search space that consists of the choice to fine-tune or freeze each module within a 12-layer VIT, where each VIT layer consists of MHSA, MLP, and LN modules. Thus our main PEFT search space $\Omega$ is the space of 36-bit binary marks $\Omega \in \{0, 1\}^{36}$. This search space contains Attention Tuning (Touvron et al., 2022), Layer Norm Tuning (Basu et al., 2023), linear readout, and full-fine tuning (Zong et al., 2023) as special cases. As ablations, we also consider 12-bit search spaces that solely search for the combination of layer-norm and attention layers to tune, as sparser alternatives to (Touvron et al., 2022; Basu et al., 2023).

**Datasets:** Our experiments include seven frequently adopted medical image analysis datasets. The selection of datasets was based on five integral factors: a) the presence of sensitive attributes, b) the presence of different potential sources of bias, c) the representation of different anatomical regions (domains), d) varying size, and e) public availability for reproducibility. Following these factors, we included Fitzpatrick17K (Groh et al., 2021; 2022), HAM10000 (Tschandl, 2018), Papila (Kovalyk et al., 2022), OL3I (Zambrano Chaves et al., 2021), OASIS-1 (Marcus et al., 2007), Harvard-GF3300 (Luo et al., 2023), and CheXpert Irvin et al. (2019). Following the settings in Zong et al. (2023), the preprocessing steps for all datasets included the binarization of the sensitive attributes (skin type, age, and sex) and the classification label along with the removal of studies with missing information. More details on data preprocessing are presented in Appendix A.4.

**Experimental Settings:** All experiments fine-tune an ImageNet-pre-trained ViT-Base model for 30 epochs with a linear warmup for 10 epochs and a cosine annealing learning rate schedular (Loshchilov & Hutter, 2016). The batch size is set to 512 and the optimizer used is AdamW (Loshchilov & Hutter, 2018). We consider one sensitive attribute at a time and note the overall performance, the worst subgroup performance and the gap between the best and worst subgroup.

**PEFT Mask Search and Hyperparameter Optimisation:** For PEFT mask search, we rely on Tree-structured Parzen Estimator (TPE) (Bergstra et al., 2011) for sampling hyperparameter values.

We also employ a pruning strategy, *successive halving* (Jamieson & Talwalkar, 2016), for early termination of unpromising trials. Our search space includes the binary mask (36 or 12 bits) along with the learning rate. For large-scale CheXpert dataset we use random sensitive-attribute balanced 10% subsampling to accelerate the HPO, and then the full train set for actual training.

Since the medical image analysis tasks are usually severely imbalanced, we use AUC rather than cross-entropy loss or accuracy as the meta-objective $\mathcal{L}^{fair}$ for the search. Since we use a gradient-free outer-loop optimizer, it is not necessary for the meta-objective to be differentiable.

As remarked in (Zong et al., 2023), existing fairness methods generally did not specify any validation criteria. They found that when conducting common fairness-driven HPO, the previously claimed differences between state-of-the-art methods vanished, and none outperformed ERM (full fine-tuning baseline, in our case). Thus we carefully ensure that all competitors optimise their learning rates based on $\mathcal{L}^{fair}$, while only our **FairTune** further optimisise the PEFT mask based on the same criterion. Our main HPO objective is disadvantaged subgroup AUC (Sagawa et al., 2020), as discussed in Sec. 3.1. We will also experiment with overall AUC as an ablation for comparison.

## 4.2 MAIN RESULTS

We report our main results in Table 1 in terms of overall test AUROC, most disadvantaged subgroup AUC, and the gap between advantaged and disadvantaged subgroups. From the table we can draw several conclusions: (1) Fine-tuning improves on both training from scratch and linear readout of the frozen features. As discussed in (Zong et al., 2023), this is a very strong baseline which many state-of-the-art purpose-designed fair learning were not able to surpass when combined with proper hyperparameter tuning. (2) Nevertheless, the state-of-the-art PEFT methods Attention Tuning (Touvron et al., 2022) and LN Tuning (Basu et al., 2023) surpass this baseline, although they are not purpose designed for fairness at all. We attribute this to their reducing the base architecture's adaptation capacity, limiting its ability to overfit to the advantaged subgroup, and thus limiting the generalisation gap (Figure 1). (3) Recent fairness interventions

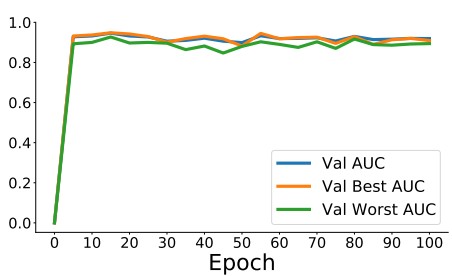

Figure 3: FairTune leads to stable fine-tuning with reduced differences between the best and worst performing subgroups compared to conventional fine-tuning from Figure 1.

FairPrune (Wu et al., 2022) and FSCL (Park et al., 2022) also underperform overall. (4) Finally, our FairTune generally achieves the best test performance overall by all metrics, with consistently good performance across all the benchmarks and sensitive attributes. (5) The second best in terms of AUC gap is training from scratch, but this corresponds to unacceptably poor performance overall, an example of the leveling-down phenomenon to be avoided (Zietlow et al., 2022).

As a qualitative illustration of how FairTune influences the fine-tuning process, we repeat the initial illustrative experiment in Figure 1, but now with the FairTune discovered mask for the same Papila dataset. From the results in Figure 3, we can see that the FairTune discovered mask leads to much fairer fine-tuning with much performance for the disadvantaged subgroup, and reduced gaps between the groups compared to the vanilla fine-tuning shown in Figure 1.

We further show in the Appendix that FairTune leads to strong performance also when using (1) alternative fairness metrics (Table 3), (2) self-supervised pre-trained ViT-Base (He et al., 2022) (Table 4), and (3) overlapping sensitive attributes (Table 5).

## 4.3 ABLATION STUDY ON THE DESIGN OF FAIRTUNE

We analyse alternative design choices for FairTune in Table 2, reporting test score average across Fitzpatrick17K, HAM10000, Papila, OL3I, OASIS-1 datasets. We first ask *What is the impact of our choice of meta-objective* $\mathcal{L}^{fair}$, as introduced in Sec 3.1? Comparing the results for FairTune (Min AUC) and FairTune (Overall AUC), we see that the min-group performance and corresponding AUC gap are clearly improved. This directly demonstrates the value of tuning model adaptation

| Dataset | Attr. | Metric | Train from Scratch | Full FT | Linear Readout | Attention Tuning | LayerNorm Tuning | FairPrune | FSCL | FairTune (Ours) |
|---|---|---|---|---|---|---|---|---|---|---|
| Fitz17k | Skin Type | Overall AUC | 73.5 | 95.9 | 90.3 | 94.1 | 92.5 | 88.4 | 89.1 | **96.7** |
| | | Min. AUC | 72.3 | 94.9 | 89.6 | 93.2 | 91.5 | 82.7 | 84.2 | **96.1** |
| | | Gap AUC | 1.6 | 3.8 | 2.8 | **1.4** | 3.5 | 6.8 | 5.6 | 2.4 |
| HAM10000 | Age | Overall AUC | 74.3 | 86.8 | 84.9 | 93.9 | 91.1 | 76.2 | 77.8 | **94.0** |
| | | Min. AUC | 66.3 | 79.2 | 75.4 | 85.9 | 83.2 | 64.3 | 67.3 | **90.1** |
| | | Gap AUC | 10.1 | 9.1 | 12.5 | 10.9 | 10.9 | 13.2 | 14.6 | **4.9** |
| | Gender | Overall AUC | 84.4 | 86.8 | 85.8 | 93.5 | 91.5 | 64.4 | 68.9 | **94.8** |
| | | Min. AUC | 83.7 | 86.3 | 85.0 | 91.9 | 90.9 | 63.9 | 66.3 | **94.4** |
| | | Gap AUC | 1.7 | 2.0 | 1.8 | 3.6 | 1.7 | 1.2 | 4.8 | **0.9** |
| Papila | Age | Overall AUC | 47.5 | 86.1 | 82.2 | 83.8 | 81.4 | 77.1 | 78.3 | **88.6** |
| | | Min. AUC | 49.3 | 81.2 | 60.7 | 78.6 | 65.2 | 71.2 | 72.7 | **85.2** |
| | | Gap AUC | **2.5** | 6.5 | 29.0 | 7.7 | 18.3 | 7.4 | 7.6 | 4.0 |
| | Gender | Overall AUC | 39.7 | 88.9 | 84.7 | 86.4 | 88.4 | 79.5 | 77.9 | **91.8** |
| | | Min. AUC | 28.9 | 88.8 | 79.5 | 80.4 | 81.0 | 74.4 | 71.4 | **90.2** |
| | | Gap AUC | 24.7 | **0.2** | 8.7 | 9.2 | 11.4 | 8.1 | 8.8 | 3.6 |
| OL3I | Age | Overall AUC | 61.9 | 67.4 | 64.4 | 64.3 | 65.3 | 66.1 | 64.4 | **72.6** |
| | | Min. AUC | 54.5 | 62.4 | 54.8 | 62.4 | 62.0 | 63.5 | 60.8 | **70.1** |
| | | Gap AUC | 7.9 | 8.4 | 14.5 | 4.7 | 5.3 | 4.4 | 5.7 | **3.6** |
| | Gender | Overall AUC | 63.8 | 65.2 | 62.8 | 74.8 | 72.9 | 65.2 | 67.6 | **78.2** |
| | | Min. AUC | 62.0 | 62.5 | 60.6 | 70.1 | 69.3 | 60.4 | 62.0 | **75.4** |
| | | Gap AUC | **2.1** | 4.0 | 2.6 | 7.5 | 4.3 | 6.5 | 8.9 | 3.7 |
| Oasis | Age | Overall AUC | 61.0 | 64.4 | 62.6 | 66.0 | 66.8 | 57.8 | 51.4 | **74.5** |
| | | Min. AUC | 57.7 | 60.1 | 59.1 | 62.2 | 60.0 | 55.6 | 50.7 | **73.2** |
| | | Gap AUC | 4.6 | 5.3 | 5.0 | 6.4 | 9.0 | 4.8 | 6.6 | **1.7** |
| | Gender | Overall AUC | 60.7 | 64.6 | 62.3 | 65.1 | 63.4 | 64.3 | 62.3 | **70.8** |
| | | Min. AUC | 56.9 | 60.3 | 59.7 | 61.5 | 61.4 | 61.8 | 60.8 | **65.5** |
| | | Gap AUC | 5.2 | 4.8 | 3.7 | 4.8 | **3.0** | 3.2 | 3.9 | 6.2 |
| Harvard-GF3300 | Age | Overall AUC | 72.3 | 82.3 | 83.6 | 81.7 | 84.7 | 74.4 | 80.4 | **86.4** |
| | | Min. AUC | 67.4 | 79.1 | 81.7 | 77.5 | 81.5 | 71.7 | 78.3 | **84.4** |
| | | Gap AUC | 6.4 | **2.5** | 3.5 | 4.8 | 3.8 | 4.8 | 3.2 | 3.2 |
| | Gender | Overall AUC | 73.4 | 80.2 | 83.4 | 80.1 | 87.6 | 74.1 | 81.1 | **88.4** |
| | | Min. AUC | 69.4 | 79.5 | 82.9 | 79.4 | 85.8 | 73.5 | 78.1 | **86.7** |
| | | Gap AUC | 7.8 | 2.8 | 2.8 | 2.9 | 2.5 | 3.1 | 2.4 | **1.9** |
| | Race | Overall AUC | 72.9 | 79.3 | 83.5 | 85.2 | 85.4 | 74.4 | 81.5 | **87.1** |
| | | Min. AUC | 67.4 | 72.7 | 80.1 | 79.7 | 80.8 | 70.6 | 76.1 | **82.4** |
| | | Gap AUC | 6.9 | 7.3 | 4.6 | 7.8 | 7.6 | 5.2 | **3.3** | 6.5 |
| CheXpert | Age | Overall AUC | 83.4 | 85.5 | 81.7 | 86.1 | 82.8 | 78.9 | 79.2 | **87.5** |
| | | Min. AUC | 78.5 | 82.3 | 77.3 | 82.3 | 79.7 | 75.6 | 77.9 | **83.8** |
| | | Gap AUC | 6.1 | 5.9 | 4.8 | 5.3 | **4.2** | **4.2** | 6.2 | 5.0 |
| | Gender | Overall AUC | 84.1 | 85.8 | 81.7 | 86.1 | 83.2 | 80.2 | 79.5 | **88.2** |
| | | Min. AUC | 81.5 | 84.1 | 80.7 | 85.0 | 80.6 | 78.5 | 77.6 | **86.5** |
| | | Gap AUC | 4.8 | **2.3** | 2.6 | 2.5 | 4.1 | 3.2 | 4.2 | 3.2 |
| | | Avg. Overall Score | 68.1 | 79.9 | 78.1 | 81.5 | 81.2 | 72.9 | 74.2 | **85.7** |
| | | Avg. Min. Score | 64.0 | 76.7 | 73.4 | 77.9 | 76.6 | 69.1 | 70.3 | **83.1** |
| | | Avg. Gap Score | 6.6 | 4.6 | 7.1 | 5.7 | 6.4 | 5.4 | 6.1 | **3.6** |
| | | Avg. Overall Rank | 7.1 | 3.5 | 5.1 | 3.3 | 3.4 | 6.5 | 6.1 | **1.0** |
| | | Avg. Min. Rank | 6.9 | 3.6 | 5.2 | 3.1 | 3.7 | 6.4 | 6.1 | **1.0** |
| | | Avg. Gap Rank | 4.9 | 4.1 | 4.4 | 5.2 | 4.7 | 4.3 | 5.5 | **2.7** |

Table 1: Evaluation of fair generalisation across medical imaging benchmarks. We report the Area Under ROC curve (AUROC) [↑, %] across the whole test set (overall) and for the most disadvantaged subgroup (min). We also report the AUROC gap [↓, %] between the advantaged and disadvantaged subgroups (Gap). All results are based on ImageNet pre-trained ViT-B, except Train from Scratch.

capacity with a fairness-specific objective rather than general purpose validation objectives. We next ask *What is the impact of our PEFT search space*, as introduced in Sec 3.2? We compare our full 36-bit search space (which includes full fine-tuning, Attention Tuning, LayerNorm tuning, and Linear Readout as special cases), with two smaller alternative 12-bit search spaces that correspond to searching for the subset of attention and layer-norm parameters to update. Between the two search spaces, AttentionTuning is better overall, but also introduces a larger AUC gap. However, the full 36-bit FairTune space is better than both of these subspaces. Nevertheless, all FairTune variants are better than the Fine-Tune baseline, in terms of Min AUC demonstrating the value of tuning model adaptation capacity. We report the full set of results in Table 6 in the Appendix.

| Metric | Fine-Tune | FairTune 12-bit, Attention Min Group AUC | FairTune 12-bit, LayerNorm Min Group AUC | FairTune 36-bit, Full Overall AUC | FairTune 36-bit, Full Min Group AUC |
|---|---|---|---|---|---|
| Overall (↑) | 78.5 | 82.4 | 81.0 | 83.5 | **84.7** |
| Min (↑) | 75.1 | 79.5 | 78.0 | 80.4 | **82.2** |
| Gap (↓) | 3.7 | 4.6 | 4.2 | 5.0 | **3.4** |

Table 2: Ablation study on FairTune design including search space $\Omega$ and objective $\mathcal{L}^{fair}$. Average test AUC over various datasets. Our 36-bit search space surpasses 12 bit alternatives. Min Group AUC as the objective leads to improved fairness compared to the conventional overall AUC.

## 4.4 ANALYSIS OF MASKS

We finally study what FairTune has learned by analysing the estimated PEFT masks, using the same subset of datasets as for our earlier analysis. We split the analysis by normalization, attention layers and MLP components. For each block, we visualize the proportion of the number of times (over the datasets and sensitive attributes) the given component was selected for fine-tuning. We further compare the masks derived from the different optimization objectives: a) Optimizing for overall performance, and b) optimizing the performance of the most disadvantaged subgroup. From the plots, we can observe that: (1) The strategies selected are non-trivial without a simple preference for either one layer type or initial vs. later layers, as expected by prior intuitively motivated work (Touvron et al., 2022; Basu et al., 2023). This demonstrates the value of automated selection of layers for updating. (2) Furthermore, the high-variability of selection for some blocks over datasets/attributes, as indicated by probabilities close to 0.5, shows the importance of *learning* dataset/attribute-specific fair tuning strategies, rather than relying on any single task-agnostic recipe. (3) The overall performance and min-subgroup performance objectives lead to substantially different masks, explaining their differing empirical performance earlier. (4) While there is substantial dataset/attribute specificity, there are some general common trends. For example, the min-subgroup objective consistently leads to freezing the first normalization layer, as well as the last four MLP layers. Meanwhile, a clear difference between the overall and min-subgroup objectives is the comparatively increased tendency of the overall objective to unfreeze the last four MLP layers.

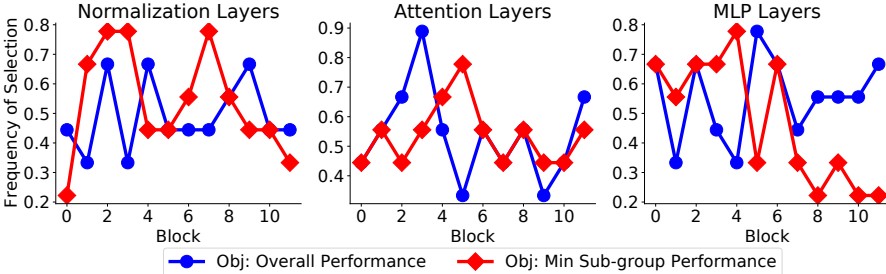

Figure 4: Frequency of selecting a specific component for fine-tuning across different scenarios.

## 5 DISCUSSION

**Potential Limitations**   The improvement in downstream fairness performance comes at a computational cost as it requires us to try various configurations of the masks, each of which corresponds to a model re-training. For example, our full FairTune pipeline takes 48GPUh on the Fitzpatrick17k dataset, compared to about 1h for unoptimized training, and 14GPUh for our HPO-tuned fine-tuning baseline. As pointed out in Zong et al. (2023), even for conventional models, proper HPO is required for optimising fairness. So the cost of a well-tuned model is inevitably much larger than a single training run. Conveniently, the cost per HPO iteration can be substantially lower in our PEFT regime, than in typical train-from-scratch HPO (Feurer & Hutter, 2019), and it could be further alleviated via using efficient techniques such as ASHA (Li et al., 2020) or PASHA (Bohdal et al., 2023) that support parallelization. Future work could also study gradient-based meta-learning (Hospedales et al., 2021) to more efficiently search higher-dimensional masks.

## 6 CONCLUSION

We provide an empirical demonstration to show that controlling the capacity of deep neural networks, particularly through the use of Parameter-Efficient Fine-Tuning methods, can lead to improved fairness on downstream tasks. Building on this finding, we introduce a framework, *FairTune*, that is fairness metric-agnostic and provides a guidance-free selection of model components to be fine-tuned. Through extensive ablation studies involving different datasets, sensitive attributes and fine-tuning strategies, we established our framework leads to consistent gains against standard fine-tuning baselines and vanilla PEFT approaches. Finally, the analysis of the selected masks has shown non-trivial scenario-dependent strategies are learned, showing the need for our proposed algorithm.

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

# A   APPENDIX

## A.1   ADDITIONAL ANALYSES

We include several further analyses, including the study of fairness transferability to other metrics, evaluation using a self-supervised pre-trained model, overlapping sensitive attributes and others.

**Impact on Other Fairness Metrics**   Our implementation of FairTune targets minAUC as the meta-objective for selecting PEFT masks that optimize this notion of fairness. One might reasonably ask how the resulting models perform in terms of other notions of fairness such as Equalized Odds Difference (EOddsD) (Agarwal et al., 2018a) and Demographic Parity Difference (DPD) (Agarwal et al., 2018a; 2019; Barocas et al., 2019). We emphasise that although these metrics are common, they have also been widely criticised in the literature for being pareto inefficient and potentially violating ethical non-maleficence (Beauchamp, 2003; Chen et al., 2018; Ustun et al., 2019; Zietlow et al., 2022). For example, it is possible to fully satisfy these criteria by providing zero-accuracy for all subgroups, which would be strictly worse than the status quo. Therefore we followed the recommendation of GroupDRO (Sagawa et al., 2020) and the recent MEDFAIR (Zong et al., 2023), and focused our evaluation on the *most disadvantaged subgroup* metric (minAUC). This metric is not vulnerable to the potential pathological outcomes that satisfy EOddsD and DPD.

Neveretheless, for completeness we evaluate our minAUC optimised models in terms of EOddsD and DPD metrics in Table 3. The results show FairTune does quite a good job of satisfying the EOddsD and DPD objectives, even though our algorithm optimises for minAUC. Where other methods outperform FairTune on these metrics, they are worse on both overall and minAUC, thus being pareto dominated by FairTune.

Finally, we remark that while we recommend minAUC objective and metric, the design of the Fair-Tune algorithm treats specific choice of optimization metric as a hyperparameter. Therefore users are easily able to plug-and-play EOddsD, or any other fairness metric as the target for FairTune to optimize (cf: Section 4.3 and Table 2).

**Combination with Other Pre-Trained Models**   Our main results are based on fine-tuning a VIT-B pre-trained by supervised learning on ImageNet. We also study the impact of fine-tuning a self-supervised pre-trained model, namely Masked AutoEncoder (MAE) (He et al., 2022). The results in Table 4 confirm that FairTune obtains both excellent overall performance and fairness, including in terms of the alternative fairness metrics.

**Overlapping Sensitive Attributes**   Our main exepriments used binary sensitive attributes. We now perform an additional experiment to further test the efficacy of *FairTune* in a scenario with multiple overlapping sensitive attributes. Specifically we used the CheXpert dataset and defined new sensitive attributes based on the intersection of the annotated *Age* and *Gender* attributes. This resulted in 4 distinct categories: (1) Patients with ages between 0 and 60 and Male gender, (2) Patients with ages 60 and above and Male gender, (3) Patients with ages between 0 and 60 and Female gender, and (4) Patients with ages 60 and above and Female gender.

The results of this experiment are presented in Table 5. We can see that FairTune leads to excellent fairness and overall performance also when using overlapping sensitive attributes. The results also indicate that good performance is consistent for supervised and self-supervised base models.

**Analysis of Outer Loop Convergence**   Figure 5 illustrates the outer loop convergence process for our datasets when using Min AUC objective on the 12-bit Attention Tuning Search Space. All datasets saturate within the given number of outer-loop iterations showing that the HPO search ran for sufficient number of trials to reach the optimal objective value.

**Full details of meta-objective comparison**   Table 6 reports the full details of the ablation study from Table 2 in the main paper.

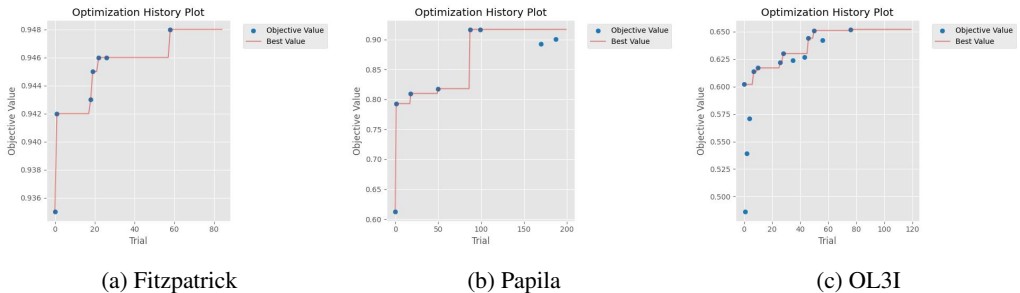

(a) Fitzpatrick            (b) Papila            (c) OL3I

Figure 5: Optimization trajectory for the outer loop PEFT mask search. The plots shown here are for the 12-bit *Attention Tuning* PEFT search space. The saturation in the objective value demonstrates that the objective is saturated within 200 outer loop iterations trials.

## A.2 DATASET DETAILS

In this section, we report the dataset details. All datasets are publicly accessible from the URLs shown in Table 7. Dataset statistics are shown in Table 8. Tables 9-13 report the specific sensitive attribute splits used for each dataset.

**Fitzpatrick17k:** We categorized the three partition labels into binary labels, specifically "benign" and "malignant." Within this categorization, we considered "non-neoplastic" and "benign" as belonging to the benign label category, while "malignant" remained in the malignant label category. Additionally, we utilized Fitzpatrick skin type labels as the sensitive attributes for our analysis.

**HAM10000:** We categorized the 7 diagnostic labels into binary labels, specifically "benign" and "malignant," in accordance with the methodology outlined by Maron et al. (2019). The "benign" category encompasses basal cell carcinoma (bcc), benign keratosis-like lesions (including solar lentigines, seborrheic keratoses, and lichen-planus like keratoses, bkl), dermatofibroma (df), melanocytic nevi (nv), and vascular lesions (comprising angiomas, angiokeratomas, pyogenic granulomas, and hemorrhage, vasc). On the other hand, the "malignant" category includes Actinic keratoses and intraepithelial carcinoma (Bowen's disease, akiec), and melanoma (mel). Images lacking recorded sensitive attributes were excluded from the dataset, resulting in a total of 9948 retained images.

**Papila:** In this dataset, we have excluded the "suspect" label class and have focused on binary classification tasks using images labeled as either "glaucomatous" or "non-glaucomatous." The dataset includes both right-eye and left-eye images of the same patients. For the purpose of splitting the dataset into training, validation, and test sets, we have followed a specific proportion, with a split of 70% for training, 10% for validation, and 20% for testing. Additionally, it is worth noting that we ensure that images from the same patient are not shared across these splits. This practice helps maintain the independence of the data subsets used for training, validation, and testing, which is crucial for evaluating the model's performance effectively.

**OL3I:** The Opportunistic L3 computed tomography slices for Ischemic heart disease risk assessment (OL3I) dataset comprises 8139 axial computed tomography (CT) slices acquired at the third lumbar vertebrae (L3) level of various individuals. The primary objective of this dataset is to develop a predictive model for determining whether an individual will receive a diagnosis of ischemic heart disease within one year following the CT scan, based on the provided labels, which represent the prognosis. In this analysis, both the sex and age of the individuals are considered sensitive attributes, taking into account potential disparities in the risk assessment related to these attributes.

**Oasis:** This dataset comprises a cross-sectional assortment of 416 individuals spanning an age range from 18 to 96 years old. For each of these individuals, the dataset includes 3 or 4 individual T1-weighted MRI scans, all acquired during single scan sessions. The participants encompass individuals who are right-handed, and the dataset includes both male and female subjects. Among the included subjects, 100 individuals who are over the age of 60 have received clinical diagnoses ranging from very mild to moderate Alzheimer's disease (AD). Furthermore, the dataset also incorporates a reliability dataset, which contains imaging data from 20 individuals who are not diagnosed with dementia. These individuals underwent a subsequent MRI session within 90 days of their ini-

tial imaging session for the purposes of assessing data reliability and consistency. The dataset is originally in 3D which has been converted to 2D by selecting the central slices from the MRI scan. We have categorized the labels into two categories by mapping the 'CDR' labels 0, 0.5, and 1 into one category and label 2.0 into the other category. While selecting the slices from MRI, we have selected 10% of the central slices for the first category and 25% in the case of the second category.

**Harvard-GF3300:** The dataset has been designed for fairness learning and contains 3300 2D and 3D retinal nerve samples from 3300 patients. The dataset is balanced in terms of racial groups and contains information on three different types of sensitive attributes: age, sex, and race. The classification label determines if a patient has glaucoma or not (binary classification). As a preprocessing step, we have binarized the sensitive attribute age and race, following the steps in Zong et al. (2023).

**CheXpert:** This dataset contains 224,316 chest radiographs of 65,240 patients. Each image can have one or more from a set of 14 labels depicting 14 different observations. For preprocessing, we have binarized the sensitive attribute age and used the "No Finding" label for training, validation and testing, as done in Zong et al. (2023).

## A.3 METRICS

The metrics employed in the study are explained below:

**AUC:** Area under the receiver operating characteristic curve (AUROC) is the standard metric for evaluation of the performance of binary classification tasks. The metric remains unaffected by the potential imbalance in class labels. Our assessment involves the computation of both the average AUC and the AUC for individual subgroups. Noteworthy emphasis is placed on the AUC gap and the worst-case AUC, serving as crucial indicators in the evaluation of group fairness and max-min fairness.

**Equalized Odds Difference (EOddsD):** This metric measures if a machine learning system works equally well on different subgroups by determining the true positive and false positive rates across different subgroups. A classifier, denoted as $h$, is deemed to satisfy equalized odds within a distribution over $(X, A, Y)$ if the prediction $h(X)$ exhibits conditional independence with respect to the sensitive attribute A, given the label Y. Agarwal et al. (2018a) define this as $E[h(X)|A = a, Y = y] = E[h(X)|Y = y]$. In our experiments, we provide Equalized Odds Difference from the *Fairlearn* package Weerts et al. (2023) that returns the larger of the true positive rate difference and false positive rate difference.

**Demographic Parity Difference (DPD):** Demographic parity, as a fairness metric, aims to guarantee that predictions made by a machine learning model remain impartial with regard to membership in a sensitive group. Simply put, achieving demographic parity signifies that the likelihood of a specific prediction is not contingent upon membership in a sensitive group. In the context of binary classification, demographic parity specifically entails maintaining equal selection rates across different groups. A classifier $h$ satisfies demographic parity under a distribution over $(X, A, Y)$ if its prediction $h(X)$ is statistically independent of the sensitive feature $A$. Agarwal et al. (2018a) define this as $E[h(X)|A = a] = E[h(X)]$. We used the *Fairlearn* package Weerts et al. (2023) for DPD implementation that reports the absolute difference between the highest and lowest selection rates $a \in A$.

A significant limitation of EOddsD and DPD metrics is that they can be trivially satisfied when the performance is 0. More broadly they do not necessitate strong performance while achieving fairness, which can often lead to undesirable performance losses. minAUC does not suffer from such limitation. That is, perfect minAUC is a *sufficient* condition for utopia (all subgroups solved perfectly), while EOddsD and DPD are only *necessary, but not sufficient* conditions.

## A.4 DATA PREPROCESSING AND EXPERIMENTS

We follow the data preprocessing steps as in Zong et al. (2023). Specifically, we created random splits of the entire dataset into training, validation and test with proportions of 80%, 10%, 10%. Next, we binarized the prediction labels and the sensitive attributes. Detailed instructions with examples are given here.

### A.4.1 Sensitive Attributes

**Sensitive Attribute Skin Type:** In relation to the sensitive attribute *skin type* we delineate two groups. The first group encompasses samples with skin types ranging from 0 to 2, while the second group comprises samples with skin types exceeding 2.

**Sensitive Attribute Age:** For the datasets HAM10000, Papila, OL3I, Oasis, Harvard-GF3300, and CheXpert, we establish two distinct categories to categorize the sensitive attribute *age*: one encompassing ages ranging from 0 to 60, and the other comprising individuals with ages exceeding 60.

**Sensitive Attribute Race:** We created two categories with 'Black' patients belonging to the first and 'Asian' or 'White' patients belonging to the second.

**Overlapping Sensitive Attributes (Age+Gender):** We created a new category of sensitive attributes by combining *Age* and *Gender*.

**Category 1:** Patients with ages between 0 and 60 and Male gender.
**Category 2:** Patients with ages 60 and above and Male gender.
**Category 3:** Patients with ages between 0 and 60 and Female gender.
**Category 4:** Patients with ages 60 and above and Female gender.

### A.4.2 Labels

**Fitzpatrick17k:** The first group includes samples labelled as *malignant*, while the second group contains samples with all other remaining labels.

**HAM10000:** We convert the original labels into two categories: benign and malignant. The former contained samples with original labels 'bcc', 'bkl', 'dermatofibroma', 'nv', 'vasc' while the latter category contained samples labelled as 'akiec', and 'mel'.

**Papila:** We used samples belonging to 'healthy' and 'glaucoma' categories while samples from the 'suspect' class was excluded.

**OL3I:** The original dataset already contains binary categories.

**Oasis-1:** The 'CDR' labels 0, 0.5 and 1 into one category and label 2.0 into the other category.

**Harvard-GF3300:** The original labels are already categorized into two categories: Glaucoma and Non-Glaucoma.

**CheXpert:** Originally, each sample in the dataset can correspond to one or more of the 14 labels. We used the 'No-Finding' label for training, validation and testing (as done in Zong et al. (2023)) since it is the only binary classification label.

## A.5 Experimental settings for Hyperparameter Optimization (HPO)

**Details of HPO Pruning:** We employed the Successive Halving Pruner provided in the Optuna package Akiba et al. (2019). Successive Halving (SH) is a bandit-based algorithm that identifies the best configuration amongst a set of configurations. The *min_resource* parameter was set as *auto* to automatically determine the number of minimum resources to be allocated to a trial, and we set the *reduction_factor* parameter to 4 (default value). This indicates that after the end of each rung, $\frac{1}{4^{th}}$ of the most promising trials would be promoted. For the remainder of the parameters (*min_early_stopping_rate* and *bootstrap_count*), we used the default value of 0.

**Details on the Parameter Sampler:** We employed the TPE sampler that uses the Tree-structured Parzen Estimator algorithm to sample hyperparameter values. Specifically, in each trial, TPE fits a Gaussian Mixture Model (GMM) to the parameter values (binary mask, in our case) that lead to the best values for the objective metric (min sub-group AUC, overall sub-group AUC, etc). Hence with each trial, we sample better parameters (mask) that lead to an improved value for the objective.

**Details on BLO using TPE and SH:** In each trial, the TPE sampler samples some values for the binary mask that is employed in fine-tuning. At the end of fine-tuning, we obtain a value for

the objective metric that is associated with the sampled values. Over numerous trials, the sampler determines which values better optimize the objective metric and gives them a higher preference during the sampling process. At the same time, Successive Halving (SH) prunes certain trials that do not show sufficient promise, thus saving both time and compute. More specifically, in our case, only $\frac{1}{4^{th}}$ of the trials (sampled mask values) are promoted for consideration in the next rung.

For the HPO search, we used a different number of trials for different datasets depending on their size. Specifically, we used 40 trials for Fitzpatrick17k, 60 trials for HAM10000, Oasis, and Harvard-GF3300, 100 trials for Papila and OL3I, and 40 trials for CheXpert. Notably, we ran HPO 10% of the total samples in CheXpert owing to its size.

**Scalability to Large Datasets:** We employed CheXpert Irvin et al. (2019) in order to test the scalability of our proposed framework to large datasets. The original size of CheXpert is around 220,000 samples. We randomly selected 10% (20,000) samples for conducting the HPO search. The mask obtained through this search was employed during fine-tuning on the entire original dataset. During subset selection, we ensured maintaining the ratios of sensitive attributes. Subsampling for HPO search has been shown to be effective previously (Shim et al., 2021; Visalpara et al., 2021) and our results reveal the same.

| Dataset | Attr. | Metric | Train from Scratch | Full FT | Linear Readout | Attention Tuning | LayerNorm Tuning | FairPrune | FSCL | FairTune (Ours) |
|---|---|---|---|---|---|---|---|---|---|---|
| Fitz17k | Skin Type | Overall AUC | 73.5 | 95.9 | 90.3 | 94.1 | 92.5 | 88.4 | 89.1 | **96.7** |
| | | Min. AUC | 72.3 | 94.9 | 89.6 | 93.2 | 91.5 | 82.7 | 84.2 | **96.1** |
| | | Gap AUC | 1.6 | 3.8 | 2.8 | **1.4** | 3.5 | 6.8 | 5.6 | 2.4 |
| | | EOddsD | 17.4 | 15.8 | 10.1 | 21.9 | **1.2** | 48.4 | 26.2 | 10.0 |
| | | DPD | 12.2 | **4.4** | 6.8 | 5.1 | 6.6 | 21.3 | 9.3 | 5.9 |
| HAM10000 | Age | Overall AUC | 74.3 | 86.8 | 84.9 | 93.9 | 91.1 | 76.2 | 77.8 | **94.0** |
| | | Min. AUC | 66.3 | 79.2 | 75.4 | 85.9 | 83.2 | 64.3 | 67.3 | **90.1** |
| | | Gap AUC | 10.1 | 9.1 | 12.5 | 10.9 | 10.9 | 13.2 | 14.6 | **4.9** |
| | | EOddsD | 11.5 | 8.6 | 11.1 | **8.3** | 59.3 | 41.2 | 26.7 | 10.1 |
| | | DPD | 23.1 | 12.9 | 11.9 | **7.2** | 12.7 | 23.1 | 16.5 | 13.2 |
| | Gender | Overall AUC | 84.4 | 86.8 | 85.8 | 93.5 | 91.5 | 64.4 | 68.9 | **94.8** |
| | | Min. AUC | 83.7 | 86.3 | 85.0 | 91.9 | 90.9 | 63.9 | 66.3 | **94.4** |
| | | Gap AUC | 1.7 | 2.0 | 1.8 | 3.6 | 1.7 | 1.2 | 4.8 | **0.9** |
| | | EOddsD | 67.3 | 86.9 | 70.6 | 65.9 | **32.1** | 58.1 | 97.2 | 38.1 |
| | | DPD | 5.6 | 1.8 | 1.3 | 1.6 | 3.3 | 4.1 | 4.1 | **1.1** |
| Papila | Age | Overall AUC | 47.5 | 86.1 | 82.2 | 83.8 | 81.4 | 77.1 | 78.3 | **88.6** |
| | | Min. AUC | 49.3 | 81.2 | 60.7 | 78.6 | 65.2 | 71.2 | 72.7 | **85.2** |
| | | Gap AUC | **2.5** | 6.5 | 29.0 | 7.7 | 18.3 | 7.4 | 7.6 | 4.0 |
| | | EOddsD | 36.1 | 50.1 | **32.5** | 62.5 | 40.2 | 57.1 | 58.4 | 34.5 |
| | | DPD | 45.1 | 41.8 | **27.2** | 40.6 | 42.8 | 39.2 | 41.3 | 30.3 |
| | Gender | Overall AUC | 39.7 | 88.9 | 84.7 | 86.4 | 88.4 | 79.5 | 77.9 | **91.8** |
| | | Min. AUC | 28.9 | 88.8 | 79.5 | 80.4 | 81.0 | 74.4 | 71.4 | **90.2** |
| | | Gap AUC | 24.7 | **0.2** | 8.7 | 9.2 | 11.4 | 8.1 | 8.8 | 3.6 |
| | | EOddsD | 20.5 | 16.7 | **12.4** | 20.8 | 33.2 | 39.2 | 43.1 | 12.5 |
| | | DPD | 10.0 | 9.1 | 6.9 | **3.5** | 9.7 | 7.3 | 8.2 | 4.9 |
| OL3I | Age | Overall AUC | 61.9 | 67.4 | 64.4 | 64.3 | 65.3 | 66.1 | 64.4 | **72.6** |
| | | Min. AUC | 54.5 | 62.4 | 54.8 | 62.4 | 62.0 | 63.5 | 60.8 | **70.1** |
| | | Gap AUC | 7.9 | 8.4 | 14.5 | 4.7 | 5.3 | 4.4 | 5.7 | **3.6** |
| | | EOddsD | 33.1 | 35.2 | 62.4 | **20.5** | 43.4 | 41.2 | 89.2 | 34.8 |
| | | DPD | 43.1 | 17.7 | 25.3 | 42.2 | **15.2** | 19.3 | 43.2 | 38.2 |
| | Gender | Overall AUC | 63.8 | 65.2 | 62.8 | 74.8 | 72.9 | 65.2 | 67.6 | **78.2** |
| | | Min. AUC | 62.0 | 62.5 | 60.6 | 70.1 | 69.3 | 60.4 | 62.0 | **75.4** |
| | | Gap AUC | **2.1** | 4.0 | 2.6 | 7.5 | 4.3 | 6.5 | 8.9 | 3.7 |
| | | EOddsD | 21.3 | 15.5 | **13.8** | 23.3 | 24.5 | 51.2 | 44.3 | 18.3 |
| | | DPD | 12.3 | 5.2 | 8.8 | 10.1 | 13.8 | 32.4 | 11.3 | **4.4** |
| Oasis | Age | Overall AUC | 61.0 | 64.4 | 62.6 | 66.0 | 66.8 | 57.8 | 51.4 | **74.5** |
| | | Min. AUC | 57.7 | 60.1 | 59.1 | 62.2 | 60.0 | 55.6 | 50.7 | **73.2** |
| | | Gap AUC | 4.6 | 5.3 | 5.0 | 6.4 | 9.0 | 4.8 | 6.6 | **1.7** |
| | | EOddsD | **41.4** | 44.7 | 43.8 | 46.8 | 98.2 | 74.3 | 98.2 | 44.7 |
| | | DPD | 41.1 | 35.6 | 43.6 | 38.2 | 45.3 | **31.3** | 38.2 | 35.0 |
| | Gender | Overall AUC | 60.7 | 64.6 | 62.3 | 65.1 | 63.4 | 64.3 | 62.3 | **70.8** |
| | | Min. AUC | 56.9 | 60.3 | 59.7 | 61.5 | 61.4 | 61.8 | 60.8 | **65.5** |
| | | Gap AUC | 5.2 | 4.8 | 3.7 | 4.8 | **3.0** | 3.2 | 3.9 | 6.2 |
| | | EOddsD | **52.2** | 58.3 | 93.2 | 63.3 | 84.2 | 71.4 | 84.2 | 58.3 |
| | | DPD | 25.0 | 31.3 | **15.8** | 18.3 | 20.7 | 22.1 | 22.3 | 20.6 |
| Harvard-GF3300 | Age | Overall AUC | 72.3 | 82.3 | 83.6 | 81.7 | 84.7 | 74.4 | 80.4 | **86.4** |
| | | Min. AUC | 67.4 | 79.1 | 81.7 | 77.5 | 81.5 | 71.7 | 78.3 | **84.4** |
| | | Gap AUC | 6.4 | **2.5** | 3.5 | 4.8 | 3.8 | 4.8 | 3.2 | 3.2 |
| | | EOddsD | 34.2 | 23.2 | 23.8 | 20.3 | 21.2 | 33.2 | 27.2 | **18.1** |
| | | DPD | 34.4 | 30.2 | 30.3 | 28.7 | 31.3 | 31.4 | 32.1 | **26.5** |
| | Gender | Overall AUC | 73.4 | 80.2 | 83.4 | 80.1 | 87.6 | 74.1 | 81.1 | **88.4** |
| | | Min. AUC | 69.4 | 79.5 | 82.9 | 79.4 | 85.8 | 73.5 | 78.1 | **86.7** |
| | | Gap AUC | 7.8 | 2.8 | 2.8 | 2.9 | 2.5 | 3.1 | 2.4 | **1.9** |
| | | EOddsD | 22.3 | 13.3 | **4.4** | 13.2 | 7.2 | 21.3 | 11.2 | 9.4 |
| | | DPD | 9.1 | 7.7 | **3.4** | 8.5 | 5.7 | 8.1 | 7.3 | 6.3 |
| | Race | Overall AUC | 72.9 | 79.3 | 83.5 | 85.2 | 85.4 | 74.4 | 81.5 | **87.1** |
| | | Min. AUC | 67.4 | 72.7 | 80.1 | 79.7 | 80.8 | 70.6 | 76.1 | **82.4** |
| | | Gap AUC | 6.9 | 7.3 | 4.6 | 7.8 | 7.6 | 5.2 | **3.3** | 6.5 |
| | | EOddsD | 18.1 | 9.3 | 14.2 | **6.2** | 10.1 | 25.1 | 12.2 | 12.2 |
| | | DPD | 17.2 | **6.2** | 11.3 | 13.4 | 11.2 | 16.0 | 11.4 | 12.3 |
| CheXpert | Age | Overall AUC | 83.4 | 85.5 | 81.7 | 86.1 | 82.8 | 78.9 | 79.2 | **87.5** |
| | | Min. AUC | 78.5 | 82.3 | 77.3 | 82.3 | 79.7 | 75.6 | 77.9 | **83.8** |
| | | Gap AUC | 6.1 | 5.9 | 4.8 | 5.3 | **4.2** | **4.2** | 6.2 | 5.0 |
| | | EOddsD | 23.3 | 23.3 | **22.1** | 28.9 | 25.4 | 51.4 | 31.2 | 23.4 |
| | | DPD | 9.1 | 7.4 | **6.6** | 8.4 | 18.2 | 32.1 | 38.2 | 17.1 |
| | Gender | Overall AUC | 84.1 | 85.8 | 81.7 | 86.1 | 83.2 | 80.2 | 79.5 | **88.2** |
| | | Min. AUC | 81.5 | 84.1 | 80.7 | 85.0 | 80.6 | 78.5 | 77.6 | **86.5** |
| | | Gap AUC | 4.8 | **2.3** | 2.6 | 2.5 | 4.1 | 3.2 | 4.2 | 3.2 |
| | | EOddsD | 13.2 | 11.4 | **9.7** | 11.3 | 19.3 | 12.3 | 13.2 | 10.1 |
| | | DPD | 4.1 | 2.4 | **1.9** | 2.1 | 11.4 | 12.4 | 9.1 | 8.3 |
| | | Avg. Overall Score | 68.1 | 79.9 | 78.1 | 81.5 | 81.2 | 72.9 | 74.2 | **85.7** |
| | | Avg. Min. Score | 64.0 | 76.7 | 73.4 | 77.9 | 76.6 | 69.1 | 70.3 | **83.1** |
| | | Avg. Gap Score | 6.6 | 4.6 | 7.1 | 5.7 | 6.4 | 5.4 | 6.1 | **3.6** |
| | | Avg. EOddD Score | 29.4 | 29.5 | 30.3 | 29.5 | 35.7 | 44.7 | 47.3 | **23.9** |
| | | Avg. DPD Score | 20.8 | 15.3 | **14.4** | 16.3 | 17.7 | 21.4 | 20.9 | 16.0 |
| | | Avg. Overall Rank | 7.1 | 3.5 | 5.1 | 3.3 | 3.4 | 6.5 | 6.1 | **1.0** |
| | | Avg. Min. Rank | 6.9 | 3.6 | 5.2 | 3.1 | 3.7 | 6.4 | 6.1 | **1.0** |
| | | Avg. Gap Rank | 4.9 | 4.1 | 4.4 | 5.2 | 4.7 | 4.3 | 5.5 | **2.7** |
| | | Avg. EOdd Diff Rank | 4.4 | 3.6 | 3.4 | 4.0 | 4.7 | 6.4 | 6.5 | **2.6** |
| | | Avg. DPD Rank | 6.9 | 3.6 | **2.7** | 3.4 | 4.9 | 5.6 | 5.7 | 3.1 |

Table 3: Evaluation of fair generalisation across medical imaging benchmarks. We report the Area Under ROC curve (AUROC) [↑, %] across the whole test set (overall) and for the most disadvantaged subgroup (min). We also report the AUROC gap [↓, %] between the advantaged and disadvantaged subgroups (Gap), Difference of Equalized Odds [↓, %, Agarwal et al. (2018a)] and Demographic Parity Difference [↓, %, Agarwal et al. (2018a; 2019); Barocas et al. (2019)]. All results are based on ImageNet pre-trained ViT-B, except Train from Scratch.

| Dataset | Attr. | Metric | Train from Scratch | Full FT | Linear Readout | Attention Tuning | LayerNorm Tuning | FairTune (Ours) |
|---|---|---|---|---|---|---|---|---|
| Fitz17k | Skin Type | Overall AUC | 73.5 | 94.7 | 85.6 | 73.6 | 92.1 | **95.3** |
| | | Min. AUC | 72.3 | 93.5 | 84.1 | 69.1 | 91.3 | **94.3** |
| | | Gap AUC | **1.6** | 4.6 | 7.8 | 6.4 | 3.7 | 4.2 |
| | | EOddsD | 17.4 | 15.3 | 26.2 | **1.3** | 9.9 | 9.8 |
| | | DPD | 12.2 | 5.4 | 4.5 | 11.0 | 5.2 | **3.5** |
| HAM10000 | Age | Overall AUC | 74.3 | 87.9 | 80.6 | 82.5 | 90.6 | **92.2** |
| | | Min. AUC | 66.3 | 80.3 | 71.8 | 69.6 | 81.7 | **85.2** |
| | | Gap AUC | 10.1 | 10.2 | 12.4 | 17.1 | 11.7 | **9.1** |
| | | EOddsD | 11.5 | 21.8 | **4.0** | 11.7 | 7.1 | 8.6 |
| | | DPD | 23.1 | 0.7 | 12.1 | 7.4 | 10.4 | **0.6** |
| | Gender | Overall AUC | 84.4 | 91.3 | 87.6 | 92.1 | 91.2 | **94.1** |
| | | Min. AUC | 83.7 | 90.9 | 86.5 | 91.1 | 90.5 | **93.2** |
| | | Gap AUC | 1.7 | **1.0** | 2.8 | 2.2 | 1.9 | 2.1 |
| | | EOddsD | 67.3 | 13.8 | 5.2 | 14.1 | 11.8 | **3.0** |
| | | DPD | 5.6 | 2.2 | 1.9 | 3.9 | 2.0 | **0.4** |
| Papila | Age | Overall AUC | 47.5 | 83.6 | 79.5 | 83.8 | 82.6 | **84.4** |
| | | Min. AUC | 49.3 | 79.5 | 71.1 | 76.3 | 77.1 | **80.1** |
| | | Gap AUC | **2.5** | 6.1 | 16.4 | 13.1 | 8.6 | 7.0 |
| | | EOddsD | 36.1 | 31.2 | **18.8** | 37.3 | 29.3 | 25.4 |
| | | DPD | 45.1 | 35.6 | **11.8** | 38.2 | 34.4 | 33.0 |
| | Gender | Overall AUC | 39.7 | 82.1 | 78.2 | 82.6 | 81.8 | **84.3** |
| | | Min. AUC | 28.9 | 75.9 | 71.4 | 79.6 | 74.8 | **82.1** |
| | | Gap AUC | 24.7 | 11.1 | 10.3 | 7.0 | 8.0 | **3.6** |
| | | EOddsD | 20.5 | 16.7 | 33.3 | 18.3 | 22.6 | **15.5** |
| | | DPD | 10.0 | 9.7 | 8.3 | **0.7** | 16.0 | 6.9 |
| OL3I | Age | Overall AUC | 61.9 | 70.6 | 60.4 | 67.5 | 72.1 | **73.6** |
| | | Min. AUC | 54.5 | 63.6 | 58.6 | 64.3 | 64.1 | **66.4** |
| | | Gap AUC | 7.9 | 8.3 | **4.2** | 5.5 | 9.1 | 7.9 |
| | | EOddsD | 33.1 | 34.9 | 26.8 | **23.2** | 60.0 | 36.1 |
| | | DPD | 43.1 | 7.6 | 32.9 | **1.6** | 45.4 | 30.0 |
| | Gender | Overall AUC | 63.8 | 71.4 | 60.4 | 62.6 | 70.1 | **74.4** |
| | | Min. AUC | 62.0 | 70.3 | 57.7 | 60.2 | 67.5 | **71.8** |
| | | Gap AUC | **2.1** | **2.1** | 4.9 | 3.7 | 5.7 | 4.5 |
| | | EOddsD | 21.3 | 19.7 | 16.6 | 13.8 | **11.0** | 11.8 |
| | | DPD | 12.3 | 9.8 | 9.3 | 13.6 | **6.4** | 8.2 |
| Oasis | Age | Overall AUC | 61.0 | 69.3 | 67.1 | 69.7 | 68.8 | **71.5** |
| | | Min. AUC | 57.7 | 53.7 | 60.5 | 57.8 | 51.1 | **65.4** |
| | | Gap AUC | **4.6** | 25.4 | 9.8 | 18.8 | 17.7 | 6.5 |
| | | EOddsD | 41.4 | 37.5 | 46.6 | 46.0 | **20.3** | 39.2 |
| | | DPD | 41.1 | 32.8 | 39.5 | **6.5** | 47.3 | 10.5 |
| | Gender | Overall AUC | 60.7 | 70.9 | 67.2 | 58.6 | 64.8 | **72.6** |
| | | Min. AUC | 56.9 | 57.8 | 55.1 | 57.4 | 58.3 | **66.4** |
| | | Gap AUC | 5.2 | 14.4 | 12.3 | **3.8** | 7.8 | 7.3 |
| | | EOddsD | 52.2 | 51.2 | 9.2 | **1.0** | 21.4 | 35.7 |
| | | DPD | 25.0 | 35.3 | 3.8 | **0.3** | 19.5 | 35.0 |
| Harvard-GF3300 | Age | Overall AUC | 72.3 | 84.5 | 75.2 | 75.5 | 85.1 | **86.1** |
| | | Min. AUC | 67.4 | 82.9 | 74.2 | 72.1 | 83.4 | **84.0** |
| | | Gap AUC | 6.4 | 7.0 | 3.7 | 4.9 | 2.9 | **0.2** |
| | | EOddsD | 34.2 | 24.9 | 19.4 | **13.8** | 19.8 | 21.2 |
| | | DPD | 34.4 | 24.4 | 33.4 | **17.4** | 29.3 | 21.3 |
| | Gender | Overall AUC | 73.4 | 85.1 | 76.1 | 71.4 | 79.1 | **86.1** |
| | | Min. AUC | 69.4 | 84.3 | 75.3 | 70.2 | 78.1 | **85.4** |
| | | Gap AUC | 7.8 | **2.0** | 2.7 | 2.9 | 3.8 | 2.6 |
| | | EOddsD | **22.3** | 73.1 | 70.8 | 86.4 | 71.0 | 45.8 |
| | | DPD | 9.1 | 3.1 | 6.1 | 4.7 | 10.4 | **2.4** |
| | Race | Overall AUC | 72.9 | 84.3 | 78.5 | 70.2 | 84.1 | **85.7** |
| | | Min. AUC | 67.4 | 80.5 | 75.6 | 65.5 | 79.1 | **83.5** |
| | | Gap AUC | 6.9 | 7.3 | 6.1 | 8.0 | 7.5 | **3.9** |
| | | EOddsD | 18.1 | 10.7 | **10.5** | 11.3 | 11.9 | 14.6 |
| | | DPD | 17.2 | 6.4 | 8.5 | **2.7** | 10.5 | 9.9 |
| CheXpert | Age | Overall AUC | 83.4 | 84.3 | 81.3 | 85.2 | 83.2 | **87.5** |
| | | Min. AUC | 78.5 | 81.5 | 76.2 | 81.5 | 80.5 | **85.1** |
| | | Gap AUC | 6.1 | 4.9 | 6.2 | 5.8 | 4.2 | **3.5** |
| | | EOddsD | 23.3 | 34.1 | 38.9 | 28.1 | 27.6 | **14.4** |
| | | DPD | 9.1 | 10.2 | 10.5 | **8.3** | 18.9 | 9.2 |
| | Gender | Overall AUC | 84.1 | 84.4 | 80.5 | 85.4 | 82.5 | **86.7** |
| | | Min. AUC | 81.5 | 83.1 | 79.1 | 84.0 | 79.4 | **85.5** |
| | | Gap AUC | 4.8 | 2.9 | 2.5 | 2.6 | 4.2 | **2.2** |
| | | EOddsD | 13.2 | 12.6 | 10.3 | 10.3 | 19.7 | **10.1** |
| | | DPD | **4.1** | 8.9 | 10.2 | 7.6 | 12.4 | 7.5 |
| | | Avg. Overall Score | 68.1 | 81.7 | 75.6 | 75.8 | 80.6 | **83.9** |
| | | Avg. Min. Score | 64.0 | 77.0 | 71.2 | 71.3 | 75.5 | **80.6** |
| | | Avg. Gap Score | 6.6 | 7.7 | 7.3 | 7.3 | 6.9 | **4.6** |
| | | Avg. EOddD Score | 29.4 | 28.4 | 24.0 | 22.6 | 24.5 | **20.8** |
| | | Avg. DPD Score | 20.8 | 13.7 | 13.8 | **8.9** | 19.2 | 12.7 |
| | | Avg. Overall Rank | 5.3 | 2.6 | 4.9 | 3.7 | 3.4 | **1.0** |
| | | Avg. Min. Rank | 5.3 | 2.9 | 4.7 | 3.8 | 3.4 | **1.0** |
| | | Avg. Gap Rank | 3.2 | 3.6 | 4.1 | 3.9 | 3.9 | **2.1** |
| | | Avg. EOdd Diff Rank | 4.5 | 4.1 | 3.1 | 3.3 | 3.4 | **2.6** |
| | | Avg. DPD Rank | 4.9 | 3.4 | 3.4 | 2.6 | 4.4 | **2.2** |

Table 4: Evaluation of fair generalisation across medical imaging benchmarks when using self-supervised pre-training (masked autoencoder – MAE). FairTune obtains the best average ranking in terms of all fairness metrics.

| Model | Metric | Full-FT | Linear-Readout | Attention-Tuning | LayerNorm-Tuning | FSCL | FairTune (Ours) |
|---|---|---|---|---|---|---|---|
| **Supervised ViT Base** | **Overall AUC** | 83.6 | 80.3 | 84.4 | 83.2 | 80.7 | **86.8** |
| | **Min AUC** | 78.5 | 74.9 | 77.4 | 76.3 | 78.1 | **79.7** |
| | **Gap** | 8.1 | 7.7 | 9.5 | 8.8 | **6.1** | 8.1 |
| | **EOddsD** | 20.1 | 15.5 | 22.9 | 22.6 | 18.3 | **12.6** |
| | **DPD** | 11.3 | **3.7** | 11.5 | 5.2 | 7.3 | 9.1 |
| **ViT Base MAE** | **Overall AUC** | 82.4 | 81.3 | 84.4 | 82.5 | 82.3 | **85.5** |
| | **Min AUC** | 79.1 | 74.9 | 77.4 | 75.1 | 79.5 | **80.1** |
| | **Gap** | 7.1 | 7.7 | 9.5 | 7.8 | 7.1 | **6.2** |
| | **EOddsD** | 21.84 | 16.2 | 23.8 | 20.4 | 19.2 | **11.4** |
| | **DPD** | 12.5 | **4.1** | 10.8 | 6.1 | 9.5 | 8.5 |

Table 5: Table showing the results for the ViT Base model pre-trained in a supervised (ImageNet) and self-supervised (MAE) fashion for the overlapping sensitive attributes in the CheXpert dataset. *FairTune* achieves the best performance in terms of the AUC (overall and min sub-group) and difference of equalized odds (EOddsD) across both the pre-training scenarios.

| Dataset | Sens Attr | Metric | FairTune: Attention Tuning (12-bit) | FairTune: LayerNorm Tuning (12-bit) | FairTune: (36-bit) Obj: Overall AUC | FairTune: (36-bit) Obj: Min Group AUC |
|---|---|---|---|---|---|---|
| Fitz17k | Skin Type | Overall | 95.9 | 94.0 | 96.1 | 96.7 |
| | | Min. | 94.9 | 93.4 | 95.2 | **96.1** |
| | | Gap | 3.8 | 2.2 | 3.2 | 2.4 |
| HAM10000 | Age | Overall | 93.9 | 93.0 | 92.8 | 94.0 |
| | | Min. | 87.6 | 88.4 | 86.1 | **90.1** |
| | | Gap | 8.5 | 5.8 | 9.3 | 4.9 |
| | Gender | Overall | 93.6 | 91.6 | 91.7 | 94.8 |
| | | Min. | 93.4 | 91.1 | 93.2 | **94.4** |
| | | Gap | 0.9 | 0.7 | 1.0 | 0.9 |
| Papila | Age | Overall | 84.2 | 88.1 | 88.2 | 88.6 |
| | | Min. | 83.3 | 84.5 | **85.4** | 85.2 |
| | | Gap | 2.2 | 4.4 | 3.0 | 4.0 |
| | Gender | Overall | 88.4 | 90.1 | 92.3 | 91.8 |
| | | Min. | 85.3 | 83.1 | 87.2 | **90.2** |
| | | Gap | 3.4 | 8.0 | 5.5 | 3.6 |
| OL3I | Age | Overall | 71.4 | 68.6 | 71.4 | 72.6 |
| | | Min. | 65.8 | 67.3 | 68.4 | **70.1** |
| | | Gap | 9.8 | 1.9 | 6.2 | 3.6 |
| | Gender | Overall | 76.8 | 72.9 | 75.6 | 78.2 |
| | | Min. | 74.8 | 70.1 | 72.0 | **75.4** |
| | | Gap | 2.7 | 4.0 | 6.1 | 3.7 |
| Oasis | Age | Overall | 69.3 | 66.5 | 73.8 | 74.5 |
| | | Min. | 67.7 | 63.1 | 72.1 | **73.2** |
| | | Gap | 2.4 | 5.0 | 1.9 | 1.7 |
| | Gender | Overall | 68.2 | 64.4 | 69.2 | 70.8 |
| | | Min. | 62.4 | 61.3 | 63.9 | **65.5** |
| | | Gap | 7.5 | 6.2 | 8.6 | 6.2 |
| | Avg. Overall Rank | | 2.8 | 3.7 | 2.3 | **1.1** |
| | Avg. Min. Rank | | 3.0 | 3.6 | 2.3 | **1.1** |
| | Avg. Gap Rank | | 2.4 | 2.3 | 3.2 | **1.8** |

Table 6: Ablation study on the design of the FairTune algorithm – full set of results. Results highlighted in bold indicate the best minimum sub-group performance for each dataset and sensitive attribute.

| Dataset | Access |
|---------|--------|
| Fitzpatrick17k | `https://github.com/mattgroh/fitzpatrick17k` |
| HAM10000 | `https://dataverse.harvard.edu/dataset.xhtml?`
`persistentId=doi:10.7910/DVN/DBW86T` |
| PAPILA | `https://www.nature.com/articles/s41597-022-01388-1#Sec6` |
| OL3I | `https://stanfordaimi.azurewebsites.net/datasets/`
`3263e34a-252e-460f-8f63-d585a9bfecfc` |
| Oasis-1 | `https://www.oasis-brains.org/#data` |
| Harvard-GF3300 | `https://ophai.hms.harvard.edu/datasets/`
`harvard-glaucoma-fairness-3300-samples/` |
| CheXpert | `https://stanfordaimi.azurewebsites.net/datasets/`
`8cbd9ed4-2eb9-4565-affc-111cf4f7ebe2` |

Table 7: URLs to access the datasets.

| Dataset | Modality | Size | Sensitive Attribute |
|---------|----------|------|---------------------|
| PAPILA | Fundus Image (2D) | 420 | Age, Gender |
| OL3I | Heart CT (2D) | 8139 | Age, Gender |
| HAM10000 | Skin Dermatology (2D) | 9948 | Age, Gender |
| OASIS | Brain MRI (3D ->2D) | 12156 | Age, Gender |
| Fitzpatrick17K | Skin Dermatology (2D) | 16012 | Skin Type |
| Harvard-GF3300 | Retinal Nerves (2D) | 3300 | Age, Gender, Race |
| CheXpert | Chest Radiographs (2D) | 222,793 | Age, Gender |

Table 8: Table representing detailed statistics of the datasets including the modality, dataset size obtained after filtering for incomplete studies, and sensitive attributes present in the metadata.

| Attribute \ Label | Label 0 | Label 1 | Split |
|-------------------|---------|---------|-------|
| Skin Type 0 | 7538 | 1312 | Train |
| Skin Type 1 | 3541 | 418 | |
| Skin Type 0 | 940 | 159 | Validation |
| Skin Type 1 | 456 | 46 | |
| Skin Type 0 | 934 | 180 | Test |
| Skin Type 1 | 443 | 45 | |

Table 9: Sensitive Attribute v/s Label Distribution for Fitzpatrick dataset.

| Attribute \ Label | Label 0 | Label 1 | Split |
|-------------------|---------|---------|-------|
| Age Group 0 | 5218 | 545 | Train |
| Age Group 1 | 1618 | 586 | |
| Age Group 0 | 604 | 69 | Validation |
| Age Group 1 | 219 | 97 | |
| Age Group 0 | 653 | 70 | Test |
| Age Group 1 | 208 | 71 | |

(a) Sensitive Attribute: Age

| Attribute \ Label | Label 0 | Label 1 | Split |
|-------------------|---------|---------|-------|
| Male | 3609 | 725 | Train |
| Female | 3224 | 406 | |
| Male | 419 | 104 | Validation |
| Female | 399 | 62 | |
| Male | 464 | 79 | Test |
| Female | 395 | 62 | |

(b) Sensitive Attribute: Gender

Table 10: Sensitive Attribute v/s Label Distribution for HAM10000 dataset.

| Attribute \ Label | Label 0 | Label 1 | Split |
|---|---|---|---|
| Age Group 0 | 3836 | 1768 | Train |
| Age Group 1 | 1716 | 2392 | |
| Age Group 0 | 468 | 312 | Validation |
| Age Group 1 | 312 | 104 | |
| Age Group 0 | 364 | 260 | Test |
| Age Group 1 | 260 | 364 | |

(a) Sensitive Attribute: Age

| Attribute \ Label | Label 0 | Label 1 | Split |
|---|---|---|---|
| Male | 1476 | 1924 | Train |
| Female | 4076 | 2236 | |
| Male | 208 | 104 | Validation |
| Female | 572 | 312 | |
| Male | 260 | 104 | Test |
| Female | 364 | 520 | |

(b) Sensitive Attribute: Gender

Table 11: Sensitive Attribute v/s Label Distribution for PAPILA dataset.

| Attribute \ Label | Label 0 | Label 1 | Split |
|---|---|---|---|
| Age Group 0 | 3512 | 87 | Train |
| Age Group 1 | 1487 | 141 | |
| Age Group 0 | 830 | 13 | Validation |
| Age Group 1 | 417 | 43 | |
| Age Group 0 | 1060 | 25 | Test |
| Age Group 1 | 478 | 46 | |

(a) Sensitive Attribute: Age

| Attribute \ Label | Label 0 | Label 1 | Split |
|---|---|---|---|
| Male | 1984 | 111 | Train |
| Female | 3015 | 117 | |
| Male | 498 | 32 | Validation |
| Female | 749 | 24 | |
| Male | 629 | 39 | Test |
| Female | 909 | 32 | |

(b) Sensitive Attribute: Gender

Table 12: Sensitive Attribute v/s Label Distribution for OL3I dataset.

| Attribute \ Label | Label 0 | Label 1 | Split |
|---|---|---|---|
| Age Group 0 | 3836 | 1768 | Train |
| Age Group 1 | 1716 | 2392 | |
| Age Group 0 | 468 | 312 | Validation |
| Age Group 1 | 312 | 104 | |
| Age Group 0 | 364 | 260 | Test |
| Age Group 1 | 260 | 364 | |

(a) Sensitive Attribute: Age

| Attribute \ Label | Label 0 | Label 1 | Split |
|---|---|---|---|
| Male | 1476 | 1924 | Train |
| Female | 4076 | 2236 | |
| Male | 208 | 104 | Validation |
| Female | 572 | 312 | |
| Male | 260 | 104 | Test |
| Female | 364 | 520 | |

(b) Sensitive Attribute: Gender

Table 13: Sensitive Attribute v/s Label Distribution for OASIS dataset.

