# OpenReview forum: "FairTune: Optimizing Parameter Efficient Fine Tuning for Fairness in Medical Image Analysis"
_ICLR.cc/2024/Conference — ICLR 2024 poster_

### Official Review · Reviewer_iDy7 · 2023-10-29

**Soundness:** 3 good
**Presentation:** 3 good
**Contribution:** 3 good
**Rating:** 6
**Confidence:** 3

**Summary:**

This paper aims to promote the group fairness of the deep learning model in medical image analysis, with a specific focus on validation fairness. The authors propose a parameter-efficient fine-tuning method to update parameters regarding fairness. The proposed method is validated on five medical imaging datasets and outperforms compared methods.

**Strengths:**

- This paper studies promoting group fairness, which is an important topic.
- The motivation is well demonstrated.
- The overall framework design is easy to follow.
- The proposed method outperforms the compared methods.

**Weaknesses:**

- In the method part, the authors limit the method fairness in binary classification. It has not mentioned how to extend the methodology for multi-classification.
- The first challenge of PEFT is related to the dataset itself, which is not a challenge for fairness.
- The method details are not clear. E.g., how to solve the BLO problem by using TPE with SH.
- It is not clear how to split the train/val/test data.
- Since this method utilizes validation data to tune the model, it is not proper to report the validation AUC; instead, test AUC should be reported.
- The experiment only validates the AUC within subgroups, more comprehensive metrics are expected (e.g., equal opportunity.)

**Questions:**

- Why the metric for fair learning is to minimize the largest loss of a subgroup instead of pursuing a uniform loss distribution among subgroups?
- How to explain the differences between masks by using different optimizing objectives?

---

> ### Author Response · Authors · 2023-11-19
> **Response (1/2)**
>
> We appreciate the effort and time spent reviewing our paper, and we are grateful that you highlight that our method performs well, its design is easy to follow and is well motivated, and focuses on important topic. We respond to the weaknesses and questions next.
>
> **Binary and multi-class classification**: We focus on binary classification problems as these are the ones typically considered in medical settings, and following the recent MEDFAIR benchmark (Zong 2023). However, FairTune trivially extends to multi-class classification if desired. The only required change is to replace Binary Cross Entropy loss with Multi-Class Cross Entropy in Eq (2) $L^{base}$, and adjust the AUC in $L^{fair}$ to multi-class AUC.
>
> **Challenge of PEFT**: We are not sure we understand the reviewer's question. Perhaps you mean that the challenge of optimising PEFT architecture is not necessarily a fairness specific challenge? We agree that PEFT architecture optimization can be conducted with the goal of maximising conventional accuracy on a dataset, rather than maximising fairness. However, as our focus is on fairness, our contribution is to introduce a methodology that enables PEFT architecture  to be optimised for fairness, and to demonstrate the empirical results that this methodology is successful. The fact that one can use related techniques to achieve other goals besides fairness does not detract from this contribution.
>
> **Details of the method**: Our method tries to optimize the mask that specifies how we perform PEFT in a way to promote fairness. We optimize either 36 or 12 element masks (36 one in the main scenario - corresponding to normalization, attention and MLP layers for each of the 12 blocks of the ViT model). We use the default settings of Tree-structured Parzen Estimator (TPE) and Successive-Halving (SH) from the Optuna bilevel optimization library. TPE is a smart Bayesian Optimisation strategy for sampling architecture configurations that are the most promising to try next. While SH maintains computational efficiency by early termination of architecture configurations that are not promising. More details of their hyperparameters are now given in the Appendix, as well as the details of how we split data into train/val/test. In all these splits we have followed MEDFAIR. We have added these further details into the revised version of the paper as part of the Appendix.
>
> **Validation AUC**: There seems to be a misunderstanding. Our main results in Table 1 and Table 2 all report Test AUC. We've updated the paper to make this even more unambiguous.

---

> ### Author Response · Authors · 2023-11-19
> **Response (2/2)**
>
> **Additional metrics & motivation for fair learning**: Thanks for the feedback. We agree that other metrics such as uniform loss or EqOpp, etc. are widely used. We emphasise that although they are common, they have also been widely criticised in the literature for being pareto inefficient and potentially violating ethical non-maleficence (Beauchamp 2003, Chen 2018, Ustun 2019, Zietlow 2022, etc). For example, it's possible to fully satisfy the EqOdds and EqOpp criteria by providing zero-accuracy for all subgroups, which would be strictly worse than the status quo. Therefore we followed the recommendation of GroupDRO (Sagawa 2019) and the recent MedFAIR (Zong 2023) and focused both our evaluation and our learning objective on the *most disadvantaged subgroup* metric (minAUC).
>
> To provide more explanation, EqOpp, EqOdds and other metrics can be satisfied by both the utopia situation (100% for all subgroups) as well as pathological situations (0% for all subgroups). Meanwhile, minAUC is not satisfied by the pathological situation, and it is only fully satisfied by the utopia situation (100% for all subgroups). Thus perfect minAUC is a sufficient condition for utopia, while the others can be sufficient but are not necessary conditions.
>
> While we prefer minAUC as a metric as explained above, for completeness we now also report other metrics - in line with requests of the other reviewers, we have decided to include Equalized Odds Difference (EOddsD) and Demographic Parity Difference (DPD). The results are in the revised paper's Table 3. We can see that FairTune does quite a good job of satisfying the EOddsD and DPD objectives, even though our algorithm optimises for minAUC. Where other methods outperform FairTune on these metrics, they are worse on both overall AUC and minAUC, thus being completely pareto dominated by FairTune.
>
> Finally, we remark that, as discussed in Sec 4.3, etc, our overall contribution and framework is agnostic to the specific meta-objective used for FairTuning. If a user for some reason really wanted to optimize for e.g. EOddsD rather than minAUC, then this is a trivial hyper-parameter to change for FairTune, which we expect would push the results towards minimising EOddsD in favour of minAUC.
>
>
> **Mask differences**: It is not easy to completely explain the specific masks discovered by different objectives. If there was a straightforward explanation for this, we could directly hand-design the masks rather than needing to automatically search for them. However, for example, Figure 4 shows that the minAUC objective prefers not to fine-tune the last few MLP layers, while the overall AUC objective does fine-tune them. This suggests that those later MLP layers are susceptible to overfitting to spurious correlations in the training set that can be detrimental to the disadvantaged subgroup within the testing set.

---

> > ### Comment · Reviewer_iDy7 · 2023-11-23
> > **Thank you for the rebuttal**
> >
> > Thanks for the rebuttal. Some of my concerns have been addressed, and below are my remaining concerns.
> >
> > **Binary and multi-class classification**:  Multi-class classification is also typical in medical settings. For example, the mentioned HAM10000 in this paper is a multi-classification dataset. Whether this method is still effective on multiple classification scenarios is underexplored. The authors are encouraged to present experimental results to support their claim that “The only required change is to replace Binary Cross Entropy loss with Multi-Class Cross Entropy, and adjust the AUC into multi-class AUC.”
> >
> >
> >
> > **Challenge of PEFT**: As the focus is on fairness, is this methodology applicable to other finetuning methods or full-finetuning?
> >
> >
> >
> > **Additional metrics & motivation for fair learning**:  What is the trivial hyper-parameter to change for FairTune? Do you mean change from AUC to EOddsD? Is it possible to minimize both metrics in the future? Some discussions on this would be helpful.

---

> > > ### Author Response · Authors · 2023-11-23
> > > **Followup response**
> > >
> > > Thank you very much for getting back to us, we provide further explanations for the given questions:
> > >
> > > **Binary and multi-class classification**: We note that binary classification remains the standard when considering fairness within medical imaging (Zong 2023), but we are very happy to do the multi-class experiments and will add them to the Appendix of the final version of the paper (as the rebuttal is expected to end very soon).
> > >
> > > **Challenge of PEFT**: Yes, our method directly applies also to other fine-tuning methods besides full fine-tuning, such as regularised fine-tuning [R1]. However, we have focused on PEFT because of its popularity and its computational efficiency, which is important for us given that we perform HPO on the fine-tuning mask.
> > >
> > > **Additional metrics & motivation for fair learning**: (1) Yes, the change would be replacing minAUC by e.g. EOddsD in the HPO objective. Note that we already demonstrated that the HPO objective can be changed in the ablation in Tab 2. So, we would simply change it again to EOddsD. (2) Yes. Multiple objectives can also be optimised simultaneously, either by summing them, or treating it as a multi-objective optimization that is supported by the Optuna HPO optimizer that we use.
> > >
> > > [R1] Xuhong Li, Yves Grandvalet, and Franck Davoine. Explicit inductive bias for transfer learning with convolutional networks, ICML’18.

---

### Official Review · Reviewer_XV7U · 2023-10-30

**Soundness:** 3 good
**Presentation:** 3 good
**Contribution:** 4 excellent
**Rating:** 6
**Confidence:** 3

**Summary:**

The paper focuses on an important field in AI, which is achieving group fairness in models, especially in medical diagnosis. They argue that this is essential but challenging due to the fairness generalisation gap where bias emerges during testing. The authors introduce a bi-level optimisation approach called FairTune, which optimises parameter-efficient fine-tuning (PEFT) techniques to balance model fit and fairness generalisation. The empirical results in the paper show that the proposed method enhances fairness across multiple medical imaging datasets.

**Strengths:**

The paper proposed a new method to finetune the pretrained model, which is potentially benefits and convenient to the current hype of foundation models or large models that require large-scale pretraining.

The proposed PEFT achieves the best performance when compared with other fairness finetuning approaches.

The paper is well written and motivates clearly as well.

**Weaknesses:**

Given the model is proposed for finetuning a pre-trained model, could the authors provide some results that using the proposed approach on finetuning Masked Autoencoder or MOCO to see if this can improve fairness for self-supervised pretraining?

For evaluation metrics in fairness, DPD and DEOdds are very common to validate an algorithm's fairness, could the authors evaluate their methods on some of the datasets using those two metrics?

The datasets compared only contains a limited number of attributes, could the authors compare their approaches to fairness medical dataset containing more sensitive attributes such as the "Luo, Yan, et al. "Harvard Glaucoma Fairness: A Retinal Nerve Disease Dataset for Fairness Learning and Fair Identity Normalization." arXiv preprint arXiv:2306.09264 (2023)."

**Questions:**

Please see weakness.

---

> ### Author Response · Authors · 2023-11-19
> **Response**
>
> We appreciate the encouraging review as well as the effort and time spent reviewing our paper. We are grateful that you highlight FairTune is clearly motivated, gives strong results and aligns well with the current interest of the ML community in foundation models. We respond to the weaknesses and questions next.
>
> **Fine-tuning of self-supervised pre-trained model**: Thanks for the suggestion. We have selected the masked autoencoder approach (MAE) for this evaluation and ran the experiments. The results confirm that FairTune still provides benefit when using self-supervised pre-training. Please see Table 4 in the revised paper.
>
> **Additional metrics**:  Thanks for the feedback. We agree that Equalized Odds Difference (EOddsD) and Demographic Parity Difference (DPD) are widely used. We emphasise that although they are common, they have also been widely criticised in the literature for being pareto inefficient and potentially violating ethical non-maleficence (Beauchamp 2003, Chen 2018, Ustun 2019, Zietlow 2022, etc). For example, it's possible to fully satisfy the EOddsD and DPD criteria by providing zero-accuracy for both subgroups, which would be strictly worse than the status quo. Therefore we followed the recommendation of GroupDRO (Sagawa 2019) and the recent MEDFAIR (Zong 2023) and focused our evaluation on the *most disadvantaged subgroup* metric (minAUC). This metric is not vulnerable to satisfaction by the potential pathological outcomes that can satisfy EOddsD and DPD.
>
> While we prefer minAUC as a metric as explained above, for completeness we now also report EOddsD and DPD in the revised paper's Table 3. We can see that FairTune does quite a good job of satisfying the EOddsD and DPD objectives, even though our algorithm optimises for minAUC. Where other methods outperform fairtune on these metrics, they are worse on both overall and minAUC, thus being completely pareto dominated by FairTune.
>
> Finally, we remark that, as discussed in Sec 4.3, etc, our overall contribution and framework is agnostic to the specific meta-objective used for FairTuning. If a user for some reason really wanted to optimize for e.g. EOddsD rather than minAUC, then this is a trivial hyper-parameter to change for FairTune, which we expect would push the results towards minimising EOddsD in favour of minAUC.
>
> **Harvard-GF3300 dataset**: Thanks for the suggestion to include this dataset, we have now evaluated FairTune and all other approaches also on this dataset and added the results into our paper. The results are consistent with our results on other datasets and show the clear benefits that FairTune brings in terms of achieving superior fairness and overall performance.

---

### Official Review · Reviewer_gkxu · 2023-10-31

**Soundness:** 3 good
**Presentation:** 3 good
**Contribution:** 3 good
**Rating:** 6
**Confidence:** 4

**Summary:**

This paper addresses the challenge of minimizing demographic bias in AI models used for medical diagnosis. The authors highlight the fairness generalization gap, where deep learning models can fit training data perfectly and exhibit fairness during training but show bias during testing when performance differs across subgroups. To tackle this issue, they propose a bi-level optimization approach called FairTune. FairTune optimizes the learning strategy based on validation fairness by adapting pre-trained models to medical imaging tasks using parameter-efficient fine-tuning techniques. The authors demonstrate empirically that FairTune improves fairness on various medical imaging datasets.

**Strengths:**

1. The paper recognizes the fairness generalization gap, where deep learning models exhibit perfect fairness during training but bias emerges during testing when generalization performance differs across subgroups.

2. This work introduce a parameter-efficient fine-tuning technique as an effective workflow for adapting pre-trained models to downstream medical imaging tasks.

3. The paper is easy to follow.

**Weaknesses:**

1. The absence of widely-used fairness metrics, such as Demographic Parity Difference [1,2,3] and Difference of Equalized Odds [1], in this work raises concerns about the completeness of the evaluation. Including these fairness metrics is essential for making the results more convincing.

2. The benchmarking presented in the study appears to be incomplete. To provide a comprehensive comparison, it is advisable to include at least two additional fairness-aware methods in the experiments: Fair Supervised Contrastive Loss [4] and Group Distributionally Robust Optimization [5].

3. Considering the relevance of MedFair [6], which evaluates fairness across various datasets, especially the significant CheXpert dataset for assessing fairness in medical applications, it would be beneficial to adhere to the experimental protocol and employ CheXpert for evaluating the proposed FairTune.

4. It is worth noting that bi-level optimization can be computationally intensive and time-consuming due to the iterative optimization required in both inner and outer loops.

5. When optimizing for fairness during fine-tuning, there is a potential concern regarding the impact on generalization performance, especially for unseen data or different subgroups. It would be valuable to clarify whether there are mechanisms in place to mitigate any adverse effects on generalization.

References:

[1] Alekh Agarwal, Alina Beygelzimer, Miroslav Dudík, John Langford, and Hanna M. Wallach. A reductions approach to fair classification. In ICML, volume 80 of Proceedings of Machine Learning Research, 60–69. PMLR, 2018.

[2] Alekh Agarwal, Miroslav Dudík, and Zhiwei Steven Wu. Fair regression: quantitative definitions and reduction-based algorithms. In ICML, volume 97 of Proceedings of Machine Learning Research, 120–129. PMLR, 2019.

[3] Solon Barocas, Moritz Hardt, and Arvind Narayanan. Fairness and Machine Learning. fairmlbook.org, 2019.

[4] Sungho Park, Jewook Lee, Pilhyeon Lee, Sunhee Hwang, Dohyung Kim, Hyeran Byun; Proceedings of the IEEE/CVF Conference on Computer Vision and Pattern Recognition (CVPR), 2022, pp. 10389-10398

[5] Sagawa, S., Koh, P. W., Hashimoto, T. B., & Liang, P. (2019, September). Distributionally Robust Neural Networks. In International Conference on Learning Representations.

[6] Zong, Y., Yang, Y., & Hospedales, T. (2022, September). MEDFAIR: Benchmarking Fairness for Medical Imaging. In The Eleventh International Conference on Learning Representations.

**Questions:**

Please refer to point 1, 2, and 3 in the weaknesses to provide more convincing empirical evidence.

Moreover, I noted that the code repository mentioned in the abstract has not been established. Providing access to the implementation code would greatly enhance the comprehensibility of this research during the review process.

----------------
Post-rebuttal comments: Thank the authors for addressing my concerns. The additional empirical evidence enhances the experiments in this work. Therefore, I would raise my rating to 'marginally above the acceptance threshold.'

---

> ### Author Response · Authors · 2023-11-19
> **Response (1/2)**
>
> We appreciate the effort and time spent reviewing our paper, and we are grateful you highlight that our work recognizes the fairness generalization gap and provides an effective workflow for adapting pre-trained models to downstream medical imaging tasks. We respond to the weaknesses and questions next.
>
> **Additional metrics**: Thanks for the feedback. We agree that Equalized Odds Difference (EOddsD) and Demographic Parity Difference (DPD) are widely used. We emphasise that although they are common, they have also been widely criticised in the literature for being pareto inefficient and potentially violating ethical non-maleficence (Beauchamp 2003, Chen 2018, Ustun 2019, Zietlow 2022, etc). For example, it's possible to fully satisfy the EOddsD and DPD criteria by providing zero-accuracy for both subgroups, which would be strictly worse than the status quo. Therefore we followed the recommendation of GroupDRO (Sagawa 2019) and the recent MEDFAIR (Zong 2023), and focused our evaluation on the *most disadvantaged subgroup* metric (minAUC). This metric is not vulnerable to satisfaction by the potential pathological outcomes that can satisfy EOddsD and DPD.
>
> While we prefer minAUC as a metric as explained above, for completeness we now also report EOddsD and DPD in the revised paper's Table 3. We can see that FairTune does quite a good job of satisfying the EOddsD and DPD objectives, even though our algorithm optimises for minAUC. Where other methods outperform FairTune on these metrics, they are worse on both overall and minAUC, thus being completely pareto dominated by FairTune.
>
> Finally, we remark that, as discussed in Sec 4.3, etc., our overall contribution and framework is agnostic to the specific meta-objective used for FairTuning. If a user for some reason really wanted to optimize for e.g. EOddsD rather than minAUC, then this is a trivial hyper-parameter to change for FairTune, which we expect would push the results towards minimising EOddsD in favour of minAUC.
>
> **Code**: We've now added the code as part of the supplementary material. Apologies for not including it as part of the initial submission.
>
> **Additional baselines**: We now add FSCL [4] into the evaluation and our results show that while this method leads to improvements over training from scratch, it is not as good as FairTune. GroupDRO was already evaluated as part of MEDFAIR, and shown to be worse than MEDFAIR's properly tuned Full FT baseline which we also use here. Furthermore, we now also compare FairTune to another recently proposed method, FairPrune [R1], that has been shown to outperform various other methods, including AdvConf [R2], AdvRev [R3], DomainIndep [R4], and OBD [R5]. Our results clearly show that FairTune outperforms both FSCL and FairPrune.

---

> ### Author Response · Authors · 2023-11-19
> **Response (2/2)**
>
> **Computational costs**: FairTune is associated with larger computational costs for the fine-tuning stage as it performs optimization on how the fine-tuning is done. In practice, while the absolute cost of FairTune is larger than conventional fine-tuning, it is not prohibitively costly, eg: 14GPUh vs 48GPUh, as discussed in Sec 5.  This is a one-off cost that allows us to make many fairer predictions. And as fairness is key in medical applications, the additional overhead is likely to be worth it. That said, we used a fairly simple HPO algorithm to expedite our research on fair fine-tuning, given the manageable absolute costs involved. There is scope for future extensions to reduce the computational cost, e.g., with more sophisticated gradient based search as discussed in various surveys (Liu et al, arXiv'21, Sinha et al, IEEE Trans Evo Comp'18, Hospedales et al, IEEE TPAMI'21).
>
> In the meanwhile, to expedite large datasets, it is possible to perform FairTune search only on a subset of the dataset to keep the costs low, before actually conducting the fine-tuning on the full dataset. Subsampling for hyperparameter optimization has been used with promising results [R6, R7] and can be used to make FairTune scalable also to these settings. Random subsampling has shown promising results already [R8] and we can extend it by randomly sampling while maintaining the ratios of sensitive attributes. We have successfully used this strategy to apply FairTune to the suggested CheXpert dataset during the short rebuttal window.
>
> **CheXpert dataset**: Thanks for the reminder. We have now added results on this benchmark, expediting FairTune's runtime by the simple heuristic of performing architecture search on a subset of the examples before fine-tuning the chosen architecture on the full dataset. The results are strong for FairTune and demonstrate that our approach is able to scale to large datasets by using subsampling.
>
> **Generalization**: Thanks for re-iterating this point. Our motivating observation from Figure 1 aligns with the reviewer's point that there is a risk of achieving good performance and good fairness on the training data, while simultaneously achieving poor generalisation on held out data. However, while many methods optimising for fairness may lead to worse generalization on held out data, our results in Table 1 suggest that our method does not suffer from it. This is because our meta-objective for hyperparameter optimization (Eq 2) *optimises for performance on held out validation data*, rather than for performance on seen training data. This is an important difference between FairTune and the vast majority of existing approaches such as GroupDRO, FairPrune, FSCL and others.
>
> References:
>
> [R1] Yawen Wu, Dewen Zeng, Xiaowei Xu, Yiyu Shi, Jingtong Hu. Fairprune: Achieving fairness through pruning for dermatological disease diagnosis. In MICCAI, 2022.
>
> [R2] Zhang, Brian Hu, Blake Lemoine, and Margaret Mitchell. Mitigating unwanted biases with adversarial learning. In AIES, 2018.
>
> [R3] Eric Tzeng, Judy Hoffman, Trevor Darrell, Kate Saenko. Simultaneous deep transfer across domains and tasks. In ICCV, 2015.
>
> [R4] Zeyu Wang, Klint Qinami, Ioannis Christos Karakozis, Kyle Genova, Prem Nair, Kenji Hata, Olga Russakovsky. Towards fairness in visual recognition: Effective strategies for bias mitigation. In CVPR, 2020.
>
> [R5] LeCun, Yann, John Denker, and Sara Solla. Optimal brain damage. In NeurIPS, 1989.
>
> [R6] Jae-hun Shim, Kyeongbo Kong, and Suk-Ju Kang. Core-set sampling for efficient neural architecture search. In ICML Workshop on Subset Selection in ML, 2021.
>
> [R7] Savan Visalpara, Krishnateja Killamsetty, and Rishabh Iyer. A data subset selection framework for efficient hyper-parameter tuning and automatic machine learning. In ICML Workshop on Subset Selection in ML, 2021.
>
> [R8] Aaron Klein, Stefan Falkner, Simon Bartels, Philipp Hennig, Frank Hutter. Fast Bayesian Optimization of Machine Learning Hyperparameters on Large Datasets. In AISTATS, 2017.

---

### Official Review · Reviewer_Dgwu · 2023-11-05

**Soundness:** 3 good
**Presentation:** 3 good
**Contribution:** 3 good
**Rating:** 6
**Confidence:** 3

**Summary:**

The paper introduces FairTune, a fine-tuning method for pre-trained models that aims to improve fairness with respect to sensitive attributes. The contribution lies in developing a technique that minimizes disparities in model performance between different demographic groups while maintaining high overall predictive accuracy. The method is demonstrated across various datasets and benchmarks, particularly in medical imaging, using the AUROC metric for evaluation.

**Strengths:**

1. FairTune provides a new pathway and improvement in reducing bias in AI models.
2. The paper conducted extensive testing over multiple datasets.
3. It leverages an ablation study to show the effectiveness of each component of the tuning process.

**Weaknesses:**

The paper may not fully address the computational costs or scalability issues associated with FairTune. Please see the questions for more details.

**Questions:**

1. The code link is not available.
2. Can the authors examine the proposed FairTune on dataset with larger "Gap"? In Table 1, the Gaps for the datasets are relatively small.  Some improvements were limited, compared with full fine-tune.
3. Can the authors provide insights into the computational overhead introduced by FairTune compared to traditional fine-tuning methods?
4. What are the scalability considerations for applying FairTune to very large datasets or models?
5. How sensitive is FairTune to the choice of sensitive attributes, and can it adapt to scenarios with multiple overlapping sensitive categories?

---

> ### Author Response · Authors · 2023-11-19
> **Response**
>
> We appreciate the encouraging review as well as the effort and time spent reviewing our paper. We are grateful that you highlight FairTune provides a new pathway and improvement in reducing bias in AI models and includes extensive evaluation. We respond to the weaknesses and questions next.
>
> **Computational costs**: FairTune is associated with larger computational costs for the fine-tuning stage as it performs optimization on how the fine-tuning is done. In practice, while the absolute cost of FairTune is larger than conventional fine-tuning, it is not prohibitively costly, eg: 14GPUh vs 48GPUh, as discussed in Sec 5.  This is a one-off cost that allows us to make many fairer predictions. And as fairness is key in medical applications, the additional overhead is likely to be worth it. That said, we used a fairly simple HPO algorithm to expedite our research on fair fine-tuning, given the manageable absolute costs involved. There is scope for future extensions to reduce the computational cost, e.g., with gradient based search as discussed in various surveys (Liu et al., arXiv'21, Sinha et al., IEEE Trans Evo Comp'18, Hospedales et al., IEEE TPAMI'21).
>
> **Scalability**: For large datasets it is possible to perform FairTune architecture search without any further algorithmic improvements, simply by performing HPO on a subset of the dataset to keep the costs low, before actually conducting the fine-tuning on the full dataset. Subsampling for hyperparameter optimization has been used with promising results [R1, R2] and can be used to make FairTune scalable also to these settings. Random subsampling has shown promising results already [R3] and we can extend it by randomly sampling while maintaining the ratios of sensitive attributes. We demonstrate this is a useful strategy by using subsampling for the search on the large CheXpert dataset, where we also demonstrate excellent results.
>
> **Code**: We've now added the code as part of the supplementary material. Apologies for not including it as part of the initial submission.
>
> **Evaluation on datasets with larger gaps**: It is important to observe that while FairTune leads to small gaps, other baselines evaluated, for example Training from scratch and Linear readout lead to large gaps of e.g. around 10 or 20 in some setups (HAM10000 - age and Papila gender especially). As a result we already evaluate scenarios that have large gaps, and we see that FairTune is able to reduce the gaps to very small values in these cases, which is a highly desirable property to have.
>
> **Sensitivity to choice of sensitive attributes and overlapping scenarios**: Our current evaluation includes 4 sets of sensitive attributes: age, gender, skin type, and race. To investigate the case of overlapping sensitive attributes, we intersected age and gender attributes using the CheXpert dataset. Our results show FairTune leads to consistent improvements even under this scenario. The details are given in the Appendix and results are shown in Table 5.
>
> References:
>
> [R1] Jae-hun Shim, Kyeongbo Kong, and Suk-Ju Kang. Core-set sampling for efficient neural architecture search. In ICML Workshop on Subset Selection in ML, 2021.
>
> [R2] Savan Visalpara, Krishnateja Killamsetty, and Rishabh Iyer. A data subset selection framework for efficient hyper-parameter tuning and automatic machine learning. In ICML Workshop on Subset Selection in ML, 2021.
>
> [R3] Aaron Klein, Stefan Falkner, Simon Bartels, Philipp Hennig, Frank Hutter. Fast Bayesian Optimization of Machine Learning Hyperparameters on Large Datasets. In AISTATS, 2017.

---

### Author Response · Authors · 2023-11-19
**Shared Response**

We thank all reviewers for taking the time to review our paper and provide feedback. Based on the feedback we have done various improvements to the paper, including:
* **Evaluation on more datasets**: Harvard-GF3300, CheXpert. The results further reinforce the excellent performance of FairTune and they also show two valuable properties: 1) consistently strong performance on datasets with a larger number of sensitive attributes, 2) scalability to large datasets via subsampling during FairTune HPO.
* **Evaluation using further approaches**: We add Fair Supervised Contrastive Loss (FSCL) and FairPrune to evaluation, and the results confirm they are not as good as FairTune. Please note that GroupDRO has already been shown to work worse than a well-tuned FullFT baseline in the MEDFAIR benchmark that has evaluated fairness on the same datasets. So we start with their well-tuned FullFT baseline, and focus on competitors not included in MEDFAIR's evaluation.
* **Evaluation with self-supervised pre-training (masked autoencoder - MAE)**: the newly added results confirm the benefits of FairTune even when using a self-supervised model as the checkpoint for fine-tuning.
* **Additional metrics**: We explain why we prefer minAUC for algorithm design and evaluation. However, for completeness we now also include Equalized Odds Difference (EOddsD) and Demographic Parity Difference (DPD), and the results show FairTune also performs well in terms of these metrics, despite that it is optimised for minAUC. This is fairly intuitive, as boosting the min group performance will also tend to improve Equalized Odds and Demographic Parity.
* **Code release and additional improvements to the paper**: we now include the code in the supplementary material and have improved the paper in various further ways e.g. more details about the methods. The updates to the paper are shown in red.

We are grateful to the reviewers for suggesting these improvements, and we believe all reviewers can now recommend acceptance for our paper. We provide further detailed answers to each reviewer separately.

---

### Meta-Review · Area_Chair_BjkD · 2023-12-11

**Metareview:**

The paper, titled "FairTune" introduces an innovative fine-tuning method targeting the reduction of demographic bias in AI models, particularly in medical diagnosis. It tackles the fairness generalization gap by proposing FairTune, a bi-level optimization approach that utilizes parameter-efficient fine-tuning techniques to adapt pre-trained models for medical imaging tasks. The paper's strengths lie in presenting FairTune as a promising avenue for bias reduction, conducting extensive testing across diverse datasets, and offering a detailed ablation study showcasing the effectiveness of each tuning component. However, notable weaknesses include a potential oversight of comprehensive considerations for computational costs or scalability issues, specific concerns about the absence of a code link, and lingering questions regarding FairTune's effectiveness on datasets with larger gaps. Reviewers express additional concerns about the computational overhead, scalability considerations, and sensitivity to sensitive attributes. Moreover, the absence of widely-used fairness metrics, incomplete benchmarking, and the need for more detailed empirical evidence, including additional fairness-aware methods and adherence to experimental protocols, are highlighted. The potential impact of bi-level optimization on generalization performance and mechanisms to mitigate adverse effects on unseen data or different subgroups are also underscored. Most of these concerns have been addressed, such as adding code, standard fairness evaluation metrics and additional datasets for evaluation. AC recommends accept.

**Justification For Why Not Higher Score:**

While the study tackles a significant problem, the meta-reviewer finds it challenging to discern the most distinctive contribution compared to existing works in terms of technical aspects. Additionally, the overall preparedness and clarity of the manuscript are lacking according to most reviewers.

**Justification For Why Not Lower Score:**

The paper introduces an innovative fine-tuning method targeting the reduction of demographic bias in AI models, particularly in medical diagnosis.  As mentioned from several reviewers (“recognizes the fairness generalization gap“, “a new pathway and improvement in reducing bias”, “extensive testing”, “easy to follow”, ”achieves the best performance”, ”an important topic”, etc.), this paper, overall, has merits that outweigh the drawbacks and should be accepted to encourage future research along this direction.

---

### Decision · Program_Chairs · 2024-01-16

Accept (poster)